# Neural Shape Compiler: A Unified Framework for Transforming between Text, Point Cloud, and Program

Tiange Luo$^{\diamond}$     Honglak Lee$^{\diamond\clubsuit\dagger}$     Justin Johnson$^{\diamond\dagger}$
$^{\diamond}$*University of Michigan*    $^{\clubsuit}$*LG AI Research*

Reviewed on OpenReview: *https://openreview.net/forum?id=gR9UVgH8PZ*

## Abstract

3D shapes have complementary abstractions from low-level geometry to part-based hierarchies to languages, which convey different levels of information. This paper presents a unified framework to translate between pairs of shape abstractions: *Text* $\Longleftrightarrow$ *Point Cloud* $\Longleftrightarrow$ *Program*. We propose **Neural Shape Compiler** to model the abstraction transformation as a conditional generation process. It converts 3D shapes of three abstract types into discrete shape code, transforms each shape code into code of other abstract types through the proposed *ShapeCode Transformer*, and decodes them to output the target shape abstraction. Point Cloud code is obtained in a class-agnostic way by the proposed *Point*VQVAE. On Text2Shape, ShapeGlot, ABO, Genre, and Program Synthetic datasets, Neural Shape Compiler shows strengths in *Text* $\Longrightarrow$ *Point Cloud*, *Point Cloud* $\Longrightarrow$ *Text*, *Point Cloud* $\Longrightarrow$ *Program*, and Point Cloud Completion tasks. Additionally, Neural Shape Compiler benefits from jointly training on all heterogeneous data and tasks.

## 1 Introduction

Humans understand 3D shapes from different perspectives: we perceive geometries, understand their composing parts and regularities, and describe them with natural language. Similarly, vision researchers designed different abstractions for 3D shapes, including (1) low-level geometric structures that plot detailed geometry such as point clouds (Gruen & Akca, 2005; Qi et al., 2017a); (2) structure-aware representations that can tell shape parts and their relations, like shape programs (Tian et al., 2019); (3) natural language to describe compositionality and functionality (Çağdaş, 1996; Chang et al., 2014). Different shape abstractions convey complementary information and encode different characteristics allowing researchers to design specialized models for various tasks (Mitra & Nguyen, 2003; Chaudhuri et al., 2020). This paper studies how to translate between *Text* $\Longleftrightarrow$ *Point Cloud* $\Longleftrightarrow$ *Program*. Such transformation mechanism can provide multi-level information about shape (Figure 1) to support various downstream tasks. Additionally, we demonstrate that combining the learning of different transformation branches within a unified framework leads to stronger representation learning and improved performance on each individual task.

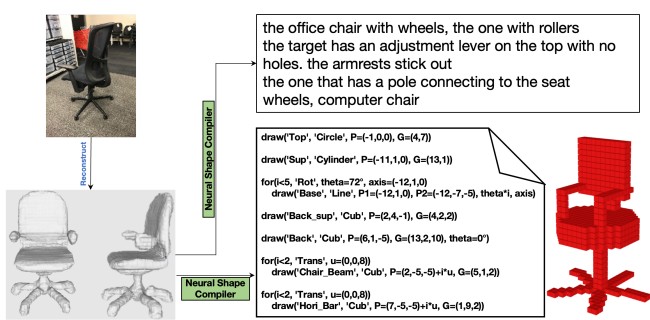

Figure 1: Example: Neural Shape Compiler helps obtain hierarchical information for the reconstruction result of a single image (Zhang et al., 2018), including its structural descriptions, regularities, and how to assemble it.

---

$^{\dagger}$Equal Advising.

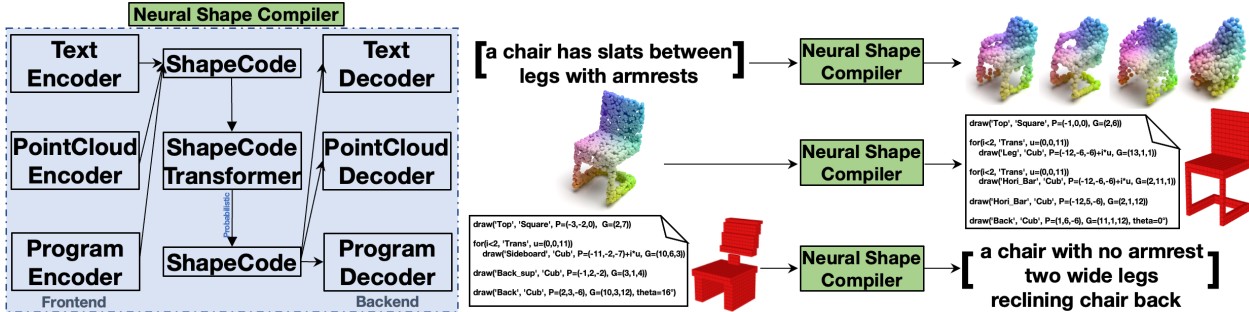

Figure 2: Overview, and three transformations conducted by Neural Shape Compiler, including *Text* ⇒ *Point Cloud*, *Point Cloud* ⇒ *Program*, and *Program* ⇒ *Text*[2]. Red shapes are the rendered results by executing the corresponding program. ShapeCode Transformer is shared across different transformations. Neural Shape Compiler is not limited to performing the transformations shown above, and benefits from joint training with all heterogeneous data and tasks.

Modern program compilers (Lattner & Adve, 2004; Zakai, 2011) turn source codes into intermediate representations (*IRs*), transform *IRs*, and decode them into other types of high-level programming languages. We take inspiration here and propose a unified architecture, ***Neural Shape Compiler***[1], to perform transformation between the three shape abstractions with neural networks. It converts all shape abstractions into discrete shape codes (i.e., IR) via their respective encoders. Then, *ShapeCode Transformer*, implemented as the standard Transformer (Vaswani et al., 2017) for generality, transforms shape codes of one type to another in an autoregressive and probabilistic manner (Ramesh et al., 2021). The probabilistic way helps us overcome the issue of ambiguity that different shape geometries can correspond to very similar program or text. Finally, the compiled shape code is decoded to the target shape abstraction through the corresponding decoder[2]. Figure 2 shows the entire process and three example transformations.

To turn point clouds with continuous coordinates into discrete shape code, we propose ***Point*VQVAE** inspired by (Van Den Oord et al., 2017). Unlike traditional point cloud process models (Qi et al., 2017a), our encoder has restricted receptive fields (Coates & Ng, 2011; Luo et al., 2019) for learning shape part embeddings. Those part embeddings let the codebook (Van Den Oord et al., 2017) encode the parts of the input 3D shape rather than the entire shape. The decoder then combines all the 3D part codes to reconstruct whole point clouds in a permutation equivariant manner.

This paper focuses on modeling 3D shape structures. Our text data contains detailed structural descriptions and is devoid of color and texture, where we adjusted and annotated data from Text2Shape (Chen et al., 2018), ShapeGlot (Achlioptas et al., 2019), and ABO (Collins et al., 2021) datasets, resulting in 107,371 (*Point Cloud*, *Structure-Related Text*) pairs. We synthesized 120,000 (*Point Cloud*, *Program*) pairs (Tian et al., 2019) for understanding shape parts and regularities with programs. Our experiments show that Neural Shape Compiler has the ability to generate point clouds from input text with geometric details, generate text descriptions of the structure of input point clouds, and generate programs for composing point clouds. Additionally, our experiments suggest the image-text pretraining model, CLIP (Radford et al., 2021), used by recent *Text* ⟹ *3D* methods (Jain et al., 2021; Sanghi et al., 2022; Nichol et al., 2022) is *NOT* suitable for generating 3D shapes with structural details.

Furthermore, we train a model using all the data from tasks *Text* ⟹ *Point Cloud*, *Point Cloud* ⟹ *Text*, and *Point Cloud* ⟹ *Program*, and train a separate limited version of the model for each task. Our experiments show that the jointly trained model consistently outperforms each limited version model on the corresponding tasks and beats existing baselines on all aforementioned tasks.

---

[1]It is important to note that Neural Shape Compiler has fundamental differences from program compilers, as described in Section 2.

[2]*Text* ⟺ *Program* is achieved in two-step generation *Text* ⟺ *Point Cloud* ⟺ *Program*. *Program* ⇒ *Point Cloud* is deterministic via executing the programs

Neural Shape Compiler is extensible. We show a case where our framework is extended to perform Point Cloud completion by transforming partial Point Cloud code into complete Point Cloud code. Limitations and future work are discussed in Section 5. Our code, dataset, and pre-trained models will be released to facilitate future research in 3D multimodal learning.

## 2 Related Work

**Compiler:** Our proposed framework is inspired by the concept of computer program compiler (Calingaert, 1979; Appel, 2004; Aho et al., 2007; Aho & Ullman, 2022). Classical compilers translate high-level program language (e.g., Pascal, C) into executable machine code with the help of assemblers and linkers (Wirth et al., 1996; Appel, 2004). Compared to classical compilers, our framework is more similar to modern program compilers (e.g., LLVM (Lattner & Adve, 2004) and Emscripten (Zakai, 2011)) where the target code is not limited to machine code and can be high-level programming languages. Our framework (Figure 2) shares similar architectures with LLVM: LLVM uses front-ends (encoders) to turn the corresponding source codes into intermediate representations (ShapeCode) and decode them via back-ends (decoders). A major difference between our framework and modern program compilers is that our IR transformation process is probabilistic, whereas the process in a program compiler is a deterministic process with potential performance optimizations.

**Multimodal Learning:** With web-scale image-text data, our community achieved remarkable progress in multi-modal learning of 2D-text (Li et al., 2019; Ramesh et al., 2021; Radford et al., 2021; Patashnik et al., 2021; Alayrac et al., 2022; Ramesh et al., 2022). However, due to the lack of large-scale 3D-text pairs and baseline systems (He et al., 2017), there is slow progress in 3D-text modeling. Some recent works tried to leverage the progress in 2D-text multimodal learning (e.g., CLIP, Imagen (Saharia et al., 2022) (Radford et al., 2021)) for 3D-text modeling (Jain et al., 2021; Zhang et al., 2021; Sanghi et al., 2022; Liu et al., 2022a; Poole et al., 2022; Lin et al., 2022; Nichol et al., 2022). However, our experiments suggest CLIP is *NOT* suitable for text-guided 3D generations once text contains structural details. Compared to them, this work studies the connections between 3D and text directly (Chen et al., 2018; Fu et al., 2022), as there are significant compositional connections between shape parts and words. Our work is closer to Text-to-Voxel works (Chen et al., 2018; Liu et al., 2022b), while none of them can generate desirable shapes corresponding to text prompt with levels of geometric details. Mittal et al. (2022) concurrently studied language-guided shape generation with learned autoregressive shape prior. Besides, the proposed framework is related to many-to-many deep learning frameworks (Ngiam et al., 2011; Jaegle et al., 2021; Reed et al., 2022).

**Generative Model:** *Point*VQVAE is inspired from VQVAE (Van Den Oord et al., 2017) studying variational auto-encoder (Kingma & Welling, 2013) with discrete latent variables (Salakhutdinov & Larochelle, 2010). We leverage its most basic concept, vector quantization (Theis et al., 2017; Agustsson et al., 2017; Oord et al., 2016), and use the discrete codes as our *IRs* in Neural Shape Compiler. We did not exhaust the complex variants or techniques for generality, such as Gumbel-softmax (Jang et al., 2016; Ramesh et al., 2021), the exponential moving average in codebook updating, and multi-scale structures (Razavi et al., 2019; Vahdat & Kautz, 2020). Prior works studied learning 3D shape probabilistic spaces with GANs for shape synthesis (Wu et al., 2016; Nguyen-Phuoc et al., 2019); (Achlioptas et al., 2018; Yang et al., 2019; Cai et al., 2020) learn the continuous distribution of point clouds and sample thereon to generate shapes; (Mittal et al., 2022; Cheng et al., 2022; Yan et al., 2022) concurrently developed ways to quantize point clouds and voxels; diffusion models are also exploited in 3D shapes (Luo & Hu, 2021; Zhou et al., 2021).

**3D Shape Understanding:** Our approach is related to research on 3D shape understanding about geometry processing and structural relationship discovery. Recent learning systems for shape processing usually perform over some low-level geometry representations, such as point clouds (Bronstein & Kokkinos, 2010; Qi et al., 2017b; Liu et al., 2019; Luo et al., 2020), volumetric grids (Wu et al., 2015; Maturana & Scherer, 2015; Wu et al., 2016; Wang et al., 2019), multi-view images (Bai et al., 2016; Feng et al., 2018; Han et al., 2019), meshes (Groueix et al., 1802; Lahav & Tal, 2020; Hu et al., 2022), and implicit functions (Mescheder et al., 2019; Park et al., 2019; Genova et al., 2020). Some represent shapes in a more structured way, including hierarchies, trees, and graphs (Wang et al., 2011; Li et al., 2017; Sharma et al., 2018; Mo et al., 2019). Neural Shape Compiler adopts point cloud as one of its shape abstractions because point cloud can describe shape structures and can be easily acquired by real sensors. Regarding the types of shape regularities (Mitra et al., 2006; Pauly et al., 2008; Mitra et al., 2007; Wang et al., 2011; Xu et al., 2012), this paper mainly considers

extrinsic part-level symmetries, including transformational, reflectional, and rotational symmetries, which relate to part pairs and multiple parts. Our framework adopts shape program (Tian et al., 2019; Jones et al., 2020; 2021; Cascaval et al., 2022), which is one of the recent advances in relationship discovery. It designed a shape domain-specific language to encode shape relationships implicitly in programs.

## 3 Method

This paper presents a unified framework that transforms between different shape abstractions and benefits from joint multimodal learning across multiple heterogeneous tasks. The key is tuning different shape abstractions into a unified discrete space, which enables our model to handle multiple tasks simultaneously. Specifically, we turn each type of input shape (*Point Cloud*, *Text*, or *Program*) into discrete shape code ($[c_1, ..., c_n], \{c_i\} \in \mathbb{Z}+$) through their corresponding encoders. ShapeCode Transformer then transforms each shape code into the code that can be decoded into target shape abstraction via the corresponding decoder. Each encoder and decoder is designated to consume the specific type of shape abstraction, while ShapeCode Transformer communicates with all encoders and decoders via discrete code for all heterogeneous tasks.

### 3.1 Encoders & Decoders

To turn point clouds $P = \{p_1, \cdots, p_m\}$ ($p_i = (x_i, y_i, z_i)$) into codes $\mathcal{C}^{point} = ([c_1^p, c_2^p, ..., c_{N_{point}}^p], \{c_i^p\} \in \mathbb{Z}+)$ while accommodating complex variations of shape structures. We draw inspiration from VQVAE (Van Den Oord et al., 2017) and propose a new model called PointVQVAE to address this task in a class-agnostic manner. Our approach involves designing a novel encoder and decoder architecture that enables PointVQVAE to reconstruct point clouds from shape codes, the same discrete space into which we convert *Text* and *Program*.

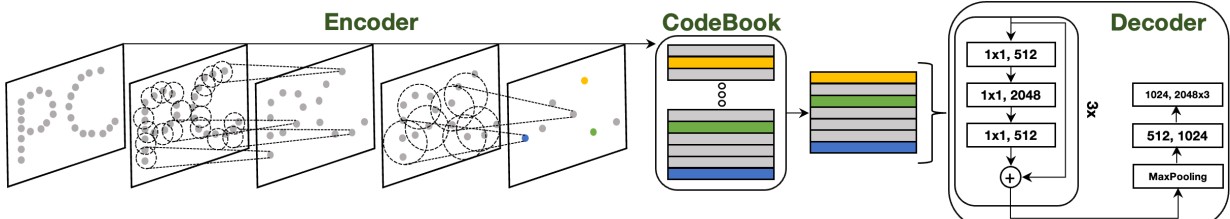

Figure 3: *Point*VQVAE consists of three parts: (1) a hierarchical encoder that has restricted receptive fields and outputs multiple part embeddings; (2) a shared codebook for vector quantization (Van Den Oord et al., 2017) (e.g., looking up the closest yellow embedding in the codebook for the yellow output of the encoder); (3) a decoder with multiple residual building blocks consisting of 1x1 convolutional layers, a max-pooling layer, and MLP layers at the end to output 2,048 points.

**Point Cloud Encoder:** The key point of *Point*VQVAE encoder is to constrain its receptive fields (Luo et al., 2016) of the last layer neurons to very local regions (Luo et al., 2019), so that the codebook behind can learn how to encode part of the 3D shape instead of entire 3D shapes (Luo et al., 2020). Specifically, we propose a hierarchical encoder with multiple down-sampling processes $T_i$ (currently implemented as a PointNet++-like structure (Qi et al., 2017b)). In each round $T_i$, we first do farthest point sampling over input point clouds to sample a set of points noted as center points. Around each center point, we draw a ball with radius $R_{T_i}$ to gather information from all points inside the query ball. After this, the input point cloud is downsampled into smaller point clouds with updated embeddings at each point. Since the radius $R_{T_i}$ of our query ball is usually smaller than the scale of the whole point clouds, the receptive field of points in downsampled point clouds can be gradually increased as we repeatedly the downsampling process (i.e., increase $\#T$). In our experiments, we set the number of rounds $\#T = 2$ and $R_{T_i}$ to be small values ($R_{T_1} = 0.1, R_{T_2} = 0.4$) for letting the final neurons (e.g., colorful points of the encoder's last layer in Figure 3) only receive local part information regarding the input shapes. Our experiments in Appendix C demonstrate that ensuring that the final neuron receives local information is critical for reconstruction.

**Point Cloud Decoder:** For each embedding in the last layer of the encoder, we look up its closest embedding in our codebook (Van Den Oord et al., 2017). We concatenate all the obtained embedding together as the input to our decoder. The concatenated embedding encodes the full information of the input shape

by collecting the information encoded by different local parts of the input shape. Compared to common point clouds auto-encoders (Qi et al., 2017b), we discard the coordinate information of the final points of the encoder. Not using coordinate information may degrade auto-encoding performance but allow us to translate between codes successfully because we do not have any coordinate information if we translate from other shape abstractions (i.e., text and program). To compensate for this point, we designed a residual-network-like block (He et al., 2016) and made our decoder with great depth to exploit the information from the concatenate embedding fully, as shown in the right of Figure 3. A max-pool layer is added to the decoder's output to ensure the final embedding is invariant to the concatenation order of the embedding from the encoder. In our experiments, we adopt Chamfer distance (Fan et al., 2017) and Earth Mover's distance (Rubner et al., 2000) as our reconstruction loss and the straight-through gradient estimator (Bengio et al., 2013) for backpropagating gradients to the encoder by copying the gradients from the inputs of the decoder to the encoder output. More details are included in Appendix B.1.

**Program Encoder & Decoder:** For program, due to the domain-specific language of shape program (Tian et al., 2019) has limited discrete types of statements $\{A, B, C, \cdots\}$ and ranges of parameters for each statement $\{a_1, a_2, \cdots, b_1 \cdots\}$, We convert the shape program into code in the order of first the statement type, then its parameters, i.e., $[A, a_1, a_2, a_3, B, b_1, b_2, b_3]$. For example, we encode the statement *draw('Top', 'Square', P=(-1,0,0), G=(2,5))* in Figure 2 as $[3, -1, 0, 0, 2, 5, 0, 0]$ where 3 represents the command of drawing square top, and the remains are its specific parameters. This way provides perfect precision to encode and decode shape programs while it cannot handle continuous parameters. Therefore, we proposed an extra ***Program*VQVAE** in Appendix B.3 to handle the case of the continuous parameters.

**Text Encoder & Decoder:** We adopt the simplest way to encode and decode text without external pre-trained parameters: leverage BPE (Sennrich et al., 2015) to encode and decode lowercase text.

## 3.2 ShapeCode Transformer

After developing all the encoders and decoders, we turn Point Clouds, Text, and Programs into the unified discrete space $\mathcal{C}^{point}$, $\mathcal{C}^{text}$, and $\mathcal{C}^{program}$, respectively. ShapeCode Transformer performs over a pair of discrete codes and transforms from one type of code to another type. Similar to (Ramesh et al., 2021), we model the pair of discrete codes as a single data stream. For example, if transforming to Point Cloud code from Text code, we model data like $[c_1^t, \cdots, c_{N_{text}}^t, c_1^p, \cdots, c_{N_{point}}^p]$. In training, we pad 0 at the left of data[3] (e.g., $[0, c_1^t, \cdots, c_{N_{text}}^t, \cdots, c_{N_{point}}^p]$) and use full attention masks (Child et al., 2019) to force ShapeCode Transformer autoregressively predicts the next token based on all the seen ones, i.e., predict $c_k$ with the context of $[0, \cdots, c_{k-1}]$. Assume $\{c_k^{gt}\}$ is the corresponding ground-truth token, our loss for this transformation is $\arg\min_\theta \sum_{k=1}^{N_{text}+N_{point}} \mathcal{L}(\mathcal{T}_\theta(0, c_1, \cdots, c_{k-1}), c_k^{gt})$, where $\mathcal{L}$ is cross-entropy loss and $\mathcal{T}$ is ShapeCode Transformer parameterized by $\theta$ to predict $c_k$ by feeding $[0, \cdots, c_{k-1}]$.

As modeled above, the second type of code gets full attention to the first type of code in predictions. This modeling can be viewed as maximizing the evidence lower bound on the joint likelihood of the distribution over the input two types of code $p_\theta(\mathcal{C}^{type_i}, \mathcal{C}^{type_j}), type \in \{point, text, program\}$, where $p_\theta$ is our ShapeCode Transformer. However, one of the challenges in our modeling is that we can generate either texts or programs conditional on the same input point clouds. *How do we let the ShapeCode Transformer know which type of object code we need?* Our solution is to encode our pairs of tokens with different positional embedding, i.e., we have two different positional encoding $\mathcal{P}_{\phi_{point}^{text}}([\mathcal{C}^{point}, \mathcal{C}^{text}])$ and $\mathcal{P}_{\phi_{point}^{program}}([\mathcal{C}^{point}, \mathcal{C}^{program}])$. Using a different positional encoding helps guide the ShapeCode Transformer to generate the specified target code.

$$\arg\min_{\theta, \phi, \omega} \sum_k \mathcal{L}(\mathcal{T}_\theta([\mathcal{P}_\phi(0), \mathcal{E}_\omega(0)], [\mathcal{P}_\phi(c_1), \mathcal{E}_\omega(c_1)], \cdots, [\mathcal{P}_\phi(c_{k-1}), \mathcal{E}_\omega(c_{k-1})]), c_k^{gt}) \qquad (1)$$

ShapeCode Transformer communicates only the shape code with all the encoders and decoders. It maintains $\mathcal{E}_\omega(c) = [\#tokens, \#dim]$ embedding for each type of code $\{\mathcal{C}^{point}, \mathcal{C}^{text}, \mathcal{C}^{program}\}$ that will be optimized during training. In each optimization, once the code is received from the encoder, ShapeCode Transformer

---

[3]Padding with 0 on the leftmost enables our framework to sample data unconditionally.

indexes the self-maintained embedding through the received code. It transforms shape code based on the indexed self-maintained embedding, enabling optimization without interference from encoder and decoder parameters. Our total training losses are the summation of equation 1 over multiple shape code pairs $c^i, c^j$ from $(\mathcal{C}^{text}, \mathcal{C}^{point})$, $(\mathcal{C}^{point}, \mathcal{C}^{text})$, and $(\mathcal{C}^{point}, \mathcal{C}^{program})$. For each input shape code, we concatenate its positional encoding $\mathcal{P}_\phi(c)$ and the embedding maintained by ShapeCode Transformer $\mathcal{E}_\phi(c)$ and feed into the ShapeCode Transfromer to predict the next token. In inference, we leverage Gumble noise (Jang et al., 2016) to generate a set of plausible results for a given input. Please refer to B.2 for all the detailed parameters.

## 4 Experiments

We conduct *Text $\Longrightarrow$ Point Cloud*, *Point Cloud $\Longrightarrow$ Text* and *Point Cloud $\Longrightarrow$ Program* tasks to verify our performance, and *Partial Point Cloud $\Longrightarrow$ Complete Point Cloud* (Shape completion) task to show the extensibility of our framework (Appendix 4.4). For each task, we have two versions of our method: Shape Compiler Limited and Shape Compiler. **Shape Compiler Limited** means that we restrict both *Point*VQVAE and ShapeCode Transformer to train on the same data and task as the baselines for fair comparisons. **Shape Compiler** is our full model trained on all tasks and data to check if joint training on heterogeneous data and tasks leads to improvement. Our used data are briefly described below. More data collection details are listed in Appendix A.

**Shape-Text Pairs:** To investigate the structural connections between text and shape, we collect a total of 107,371 (*Point Cloud, Structure-Related Text*) pairs from a total of 20,355 shapes with 9.47 words per description on average and 26,776 unique words. We use 85% shapes for training and the remaining 15% for testing. Data collection details are listed in Appendix A.

**3D Shape Assets**: To achieve a general framework, we collect various 3D shapes and obtain a total of 140,419 shapes of 144 categories from ShapeNet (Chang et al., 2015), ABO (Collins et al., 2021), and Program Synthetic (Tian et al., 2019) datasets. For each shape, we sample 10,000 points as inputs and remove all artificial colors and textures to focus on shape geometry and compositionality.

**Shape-Program Pairs**: We followed (Tian et al., 2019) and synthesized 120,000 (*Point Cloud, Program*) pairs. For testing, Tian et al. (2019) sampled shapes from ShapeNet, and we use the same sets to evaluate in our experiments for fair comparisons.

### 4.1 *Text $\Longrightarrow$ Point Cloud*

Neural Shape Compiler can generate point clouds corresponding to text prompt (Figure 4). We compare methods (Jain et al., 2021; Sanghi et al., 2022) that rely primarily on CLIP Radford et al. (2021) to generate 3D shapes and methods that require 3D-text pairs (Chen et al., 2018; Liu et al., 2022b). We use our shape-text training pairs to train CWGAN (Chen et al., 2018) and Shape IMLE diversified model (Liu et al., 2022b), and our Shape Compiler Limited for fair comparisons. Since this paper focuses on geometry, we only compare with (Chen et al., 2018; Liu et al., 2022b) in geometrical aspects, which is the same comparison way used in (Mittal et al., 2022). We compare CLIP-Forge (Sanghi et al., 2022) and DreamField (Jain et al., 2021) to show the gap between exploiting the

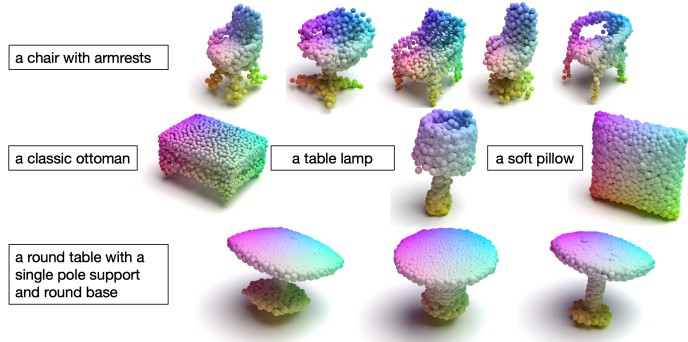

Figure 4: *Text $\Longrightarrow$ Point Cloud* by Neural Shape Compiler. A text prompt may translate to multiple shapes.

2D-text model and training over 3D-text pairs. Since DreamField learns a neural radiance field over a single text with a lot of sampling, its training time is too long to test in the scale of our test set (details in Appendix B.4). We only compare its qualitative results. In order to benchmark *Text $\Longrightarrow$ Point Cloud* task, we turn models which output 3D voxels into point clouds by sampling 2,048 points on their output voxels. Shape

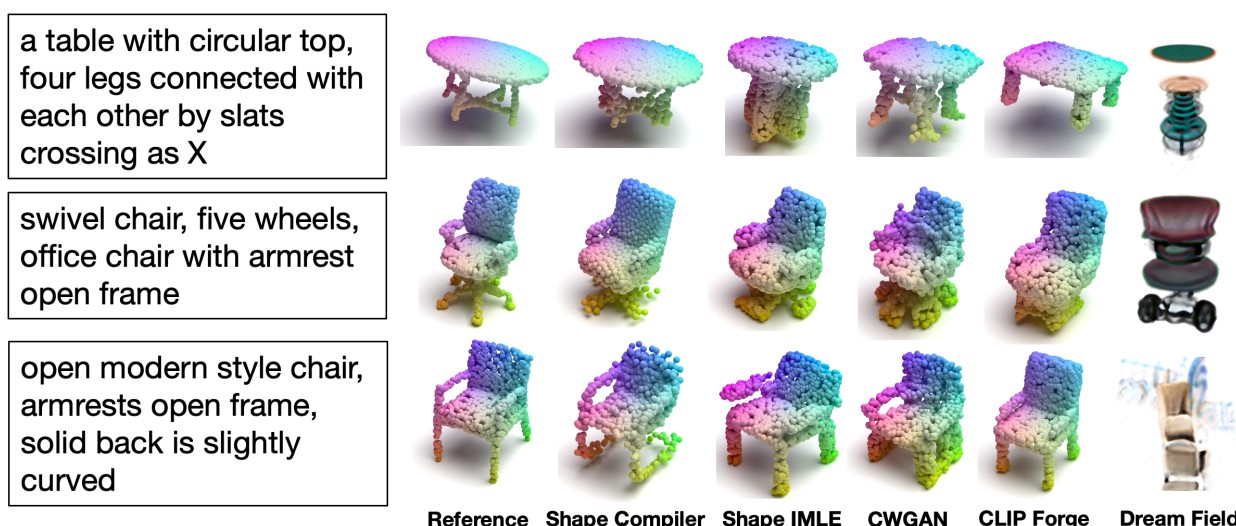

**a table with circular top, four legs connected with each other by slats crossing as X**

**swivel chair, five wheels, office chair with armrest open frame**

**open modern style chair, armrests open frame, solid back is slightly curved**

Reference   Shape Compiler   Shape IMLE   CWGAN   CLIP Forge   Dream Field

Figure 5: *Text ⟹ Point Cloud*. Neural Shape Compiler can generate corresponding point clouds to the text prompt with structural details, while the baselines generate inaccurate structures or fewer alignments with the text prompt. We visualize Point Clouds via Misuba (Jakob et al., 2022)

Compiler is trained with all heterogeneous data and tasks described in Section 4 to show the benefits from jointly training over all tasks.

**Evaluation Protocol:** To benchmark *Text ⟹ Point Cloud* task, we measure generative performance from multiple angles: quality, fidelity, and diversity. **(1)** For measuring quality, we use Minimal Matching Distance (MMD) (Achlioptas et al., 2018) to check if the generated shape distributions are close to ground-truth shape distributions by computing the distance between the set of all our generated point clouds and the set of all ground-truth point clouds. We use Chamfer distance for each pair of point clouds in computing MMD. **(2)** For measuring fidelity, we compute the minimal Chamfer distance between all generated point clouds and the corresponding ground truth shape of the input text. We report the Average Minimal Distance (AMD) over all the test texts. **(3)** For measuring diversity, we adopt Total Mutual Difference (TMD) (Wu et al., 2020), which computes the difference between all the generated point clouds of the same text inputs. For each generated point cloud $P_i$, we compute its average Chamfer distance $CD_{P_i}$ to other $k-1$ generated point clouds $P_{j_{j \neq i}}$ and compute the average: $TMD = \text{Avg}_{i=1}^{k} CD_{P_i}$. Given an input text, every model finalizes 48 normalized point clouds of 2,048 points.

**Results:** According to Table 1, Shape Compiler variants generally outperform CLIP-Forge (Sanghi et al., 2022), CWGAN (Chen et al., 2018), and Shape IMLE Diversified (Liu et al., 2022b), especially in AMD (fidelity), which demonstrates our proposed method can generate shapes that more closely match the input texts which contain levels of geometrical details. This is consistent with the qualitative results in Figure 5, where all compared baselines cannot handle the text prompt containing many structural details well. One potential drawback in (Chen et al., 2018; Liu et al., 2022b) is the use of 3D convolutional layers, which make their models tend to overfit training data distribution. This is also shown in Figure 5, where their results are less correlated with the input texts. Shape IMLE outperforms CWGAN, consistent with the discovery in (Liu et al., 2022b). However, its inference is much slower than Shape Compiler due to the use of 3D convolutional layers and implementations, whereas

| Dataset | Method | MMD ↓ | AMD ↓ | TMD ↑ |
|---------|--------|-------|-------|-------|
| ShapeGlot | CWGAN | 22.46 | 27.32 | 0.67 |
| | Shape IMLE | 12.37 | 14.92 | 1.83 |
| | CLIP-Forge | 36.43 | 64.5 | 2.45 |
| | Shape Compiler Limited | 6.19 | 11.27 | 1.85 |
| | Shape Compiler | **5.7** | **7.31** | **2.8** |
| Text2shape | CWGAN | 10.42 | 18.21 | 0.79 |
| | Shape IMLE | 6.73 | 15.34 | 2.34 |
| | CLIP-Forge | 5.16 | 19.69 | 1.34 |
| | Shape Compiler Limited | 6.21 | 14.21 | 1.53 |
| | Shape Compiler | **4.53** | **11.66** | **3.07** |
| ABO | CWGAN | 12.74 | 22.31 | 0.52 |
| | Shape IMLE | 6.43 | 13.67 | **1.42** |
| | CLIP-Forge | 7.89 | 23.35 | 1.23 |
| | Shape Compiler Limited | 4.93 | 8.33 | 0.67 |
| | Shape Compiler | **4.76** | **7.77** | 1.34 |

Table 1: *Text ⟹ Point Cloud*. MMD (quality), AMD (fidelity), and TMD (diversity) are multiplied by $10^3$, $10^3$ and $10^2$, respectively.

our method infers low-dimensional codes and decodes them
through a 1D residual decoder. In our same single GPU, Shape IMLE takes 1003.23 seconds to generate 48 shapes, while Shape Compiler takes only 14.69 seconds. Shape Compiler consistently outperforms Shape Compiler Limited indicating the proposed framework benefits from the joint training on all heterogeneous data and tasks. However, due to the use of point cloud representation, the inherent difficulty of 3D shape generation with structure details, and the introduction of Gumbel noise, our framework produces rough boundaries, uneven surfaces, and irrelevant shapes. There is a lot of room for research in the proposed framework and *Text ⟹ Point Cloud* task.

Despite the success of using CLIP in 3D shape generation with text prompt containing simple structure descriptions (Sanghi et al., 2022; Jain et al., 2021), CLIP-Forge shows undesirable AMD and MMD scores in both ShapeGlot and ABO datasets, and DreamField fails to generate structures corresponding to the text prompt, although they are perceptually fine. Additionally, our experimental results suggest that **the text embedding of CLIP models (Radford et al., 2021) is *Not* able to reflect differences in small changes in the text, which may lead to large changes in structure**. Table 2 shows several examples. The texts in the second column are similar to those in the first but will lead to significant structural changes. However, they have very high similarity scores in CLIP text embeddings, almost 1, which is

| Text1 | Text2 | Similarity |
|---|---|---|
| a chair with armrests | a chair with no armrests | 0.983 |
| a chair with armrests | a chair without armrests | 0.988 |
| a swivel chair with armrests | a swivel chair with no armrests | 0.979 |
| a swivel chair with armrests | a swivel chair without armrests | 0.986 |
| a chair with armrests | a chair with armrests and curved back | 0.962 |
| a chair with armrests | a chair with no armrests and curved back | 0.965 |
| a swivel chair with armrests | a swivel chair with armrests and curved back | 0.981 |
| a swivel chair with armrests | a swivel chair with no armrests and curved back | 0.982 |
| a table with circular top | a table with square top | 0.932 |
| a table with circular top | a table with rectangular top | 0.938 |
| a stool chair with two legs | a stool chair with three legs | 0.992 |
| a couch with two seats | a couch with three seats | 0.986 |

Table 2: The rightmost column is the cosine similarity between the CLIP text embeddings (Radford et al., 2021) of the left two texts. Cosine Similarity $= \frac{f_1 \cdot f_2}{\max\left(\|f_1\|_2 \cdot \|f_2\|_2, \epsilon\right)}$, where $f_1$ and $f_2$ are the CLIP embedding of Text1 and Text2, respectively. $\epsilon$ is 1e-6 to avoid division by zero.

an upper bound for cosine similarity. This may illustrate why CLIP-Forge (Sanghi et al., 2022) achieved low AMD scores (Table 1), where our 3D-text datasets contain complex structural descriptions as shown in Appendix A. CLIP text embeddings cannot distinguish text with structural details well, which hurts the performance of text-guided 3D shape generation. The clip model we used here is *ViT-B/32*, and other CLIP models show similar trends.

Also, we found **the cosine similarity between the CLIP embedding of rendered 3D shape images and the CLIP embedding of text cannot accurately measure the degree of alignment between the text prompt and 3D geometry when the text prompt and 3D geometry contain structural details**.

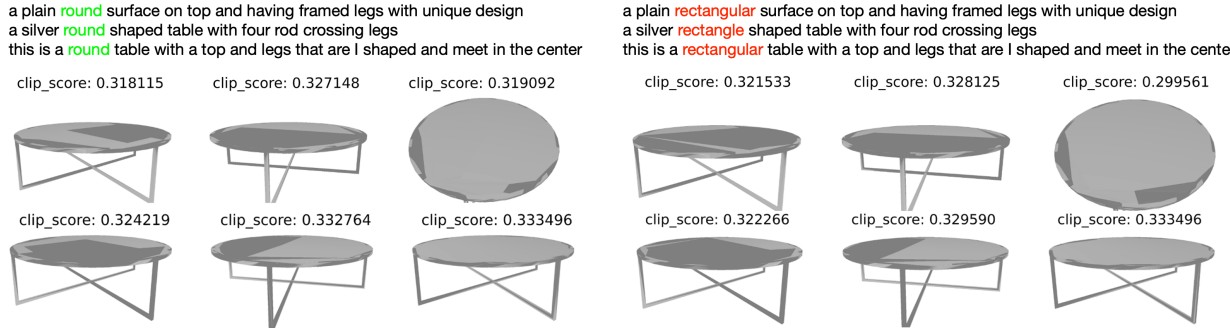

Figure 6: Render the shape with six different camera positions and two different point lights by Pytorch3D Ravi et al. (2020). The texts shown at the top left are the shape-associated texts provided in our datasets, which only contain Round or Circular. We modified them into Rectangle or Rectangular as shown in the top right. The number above each image is the average cosine similarity between each CLIP image embedding and all corresponding CLIP text embedding.

We render 3D shapes into images with six camera positions and two different point lights with PyTorch3D (Ravi et al., 2020), as shown in Figure 6. Then, we obtain each image embedding by using the CLIP image branch encoder. Each shape has several text descriptions. We tokenize those texts with the tokenizer provided by CLIP and obtain our text embedding by feeding them into the CLIP text encoder. In Figure 6, the score above each shape is the average cosine similarity between the image and each text shown at the top. We select shapes with text containing Round or Circular and not including Rectangle and Rectangular. We compute the cosine similarity between the embedding of the rendered images and two different sets of text embedding: **(1)** the original ground-truth texts which contain Round or Circular; **(2)** we replace Round or Circular into Rectangle and Rectangular. An example of replacement is shown in Figure 6 where the right figure top texts are the replaced ones. We compute the similarity scores over 1,358 shapes with a total of 6,697 texts. As described above, we have six images for each shape, resulting in 40,182 scores. We calculate the mean and standard deviation of all scores for the two sets of text: (1) $0.2925 \pm 0.02173$ and (2) $0.2917 \pm 0.02211$. Although there are huge differences between the two sets of texts, their average cosine similarity with the CLIP image embedding is very close. This experiment further supports CLIP is *NOT* suitable to deal with text containing geometric details.

### 4.2 *Point Cloud $\Longrightarrow$ Text*

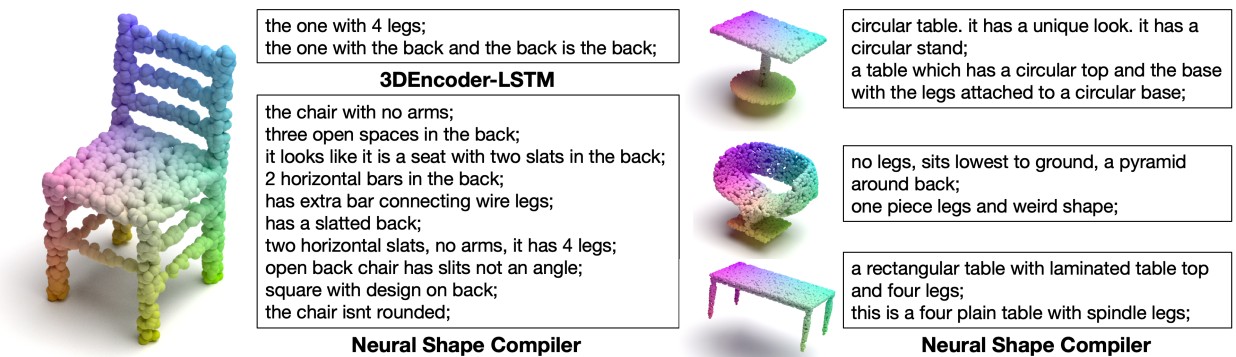

Figure 7: *Point Cloud $\Longrightarrow$ Text.* One description per sentence. The shown shapes are from test sets, and Neural Shape Compiler tell their structures well.

Neural Shape Compiler performs *Point Cloud $\Longrightarrow$ Text* tasks that describe given shape point clouds as shown in Figure 7. To compare, we followed (Chen et al., 2021) and implemented a 3DEncoder-LSTM model as our baseline by replacing the image encoder of Show-Attend-Tell(Vinyals et al., 2015) with a 3D encoder. We adopt a similar encoder structure as *Point*VQVAE and set the output feature to be 512 dimension, the same value as our intermediate representations. An LSTM model will then be trained over the 512 output features to predict words gradually, and an end word will be the last token. For each shape, all methods will generate 48 descriptions to compare.

**Evaluation Protocol:** We measure the quality of the generated shape descriptions with our annotations in validation sets. Here we adopt the common metrics used in text generations tasks (Vinyals et al., 2015; Koncel-Kedziorski et al., 2019) including BLEU (4-gram) (Papineni et al., 2002), CIDER (Vedantam et al., 2015), and ROUGE (Lin, 2004). We also adopt Dist-1 and Dist-2 in (Li et al., 2015) to measure the diversity of the generated results: Dist-1 and Dist-2 are the results of distinct unigrams and bigrams divided by the total number of tokens (maximum 100 in our table).

| Dataset | Method | BLEU-4 ↑ | CIDER ↑ | ROUGE ↑ | Dist-1 ↑ | Dist-2 ↑ |
|---|---|---|---|---|---|---|
| ShapeGlot | 3DEncoder-LSTM | 2.78 | 2.93 | 18.61 | 0.016 | 0.017 |
| | Shape Compiler Limited | 2.96 | 2.15 | 21.87 | 1.174 | 14.146 |
| | Shape Compiler | **3.19** | **2.35** | **22.21** | 1.193 | 14.784 |
| | Ground-Truth | - | - | - | 6.908 | 37.92 |
| Text2Shape | 3DEncoder-LSTM | 1.01 | 0.84 | 21.18 | 0.027 | 0.042 |
| | Shape Compiler Limited | 2.96 | 2.21 | 25.97 | 0.792 | 9.984 |
| | Shape Compiler | **4.25** | **2.96** | **27.61** | 0.921 | 11.343 |
| | Ground-Truth | - | - | - | 5.906 | 34.05 |
| ABO | 3DEncoder-LSTM | 0.01 | 1.39 | 2.98 | 0.031 | 0.047 |
| | Shape Compiler Limited | **8.95** | 106.31 | 25.42 | 3.605 | 23.165 |
| | Shape Compiler | 7.98 | **111.22** | **26.47** | 1.27 | 13.43 |
| | Ground-Truth | - | - | - | 18.294 | 49.097 |

Table 3: *Point Cloud $\Longrightarrow$ Text.* BLEU, CIDER, and ROUGE measure quality. Dist-1 and Dist-2 measure diversity.

**Results:** According to Dist-1 and Dist-2 in Table 3, Shape Compiler generates much more

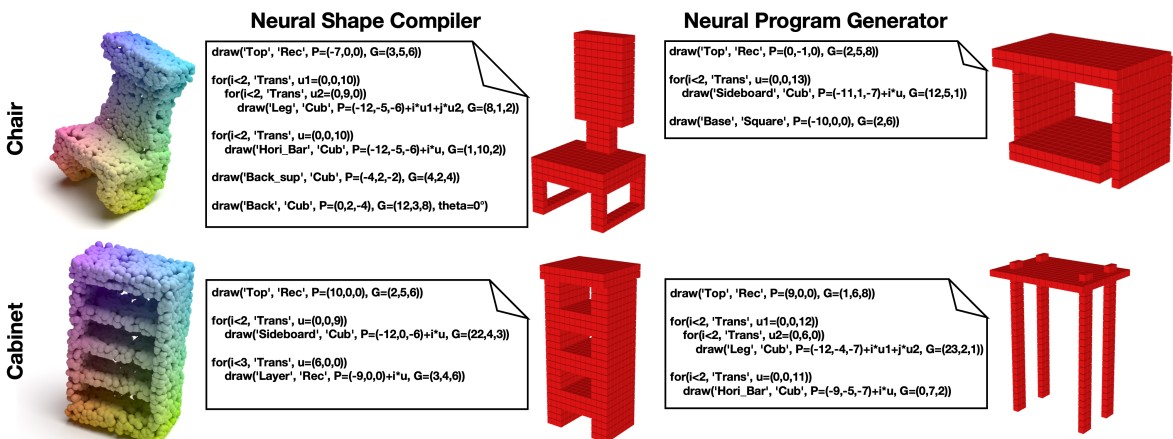

Figure 8: *Point Cloud ⟹ Program.* Neural Shape Compiler can well infer programs for the shown shapes of unseen categories and reconstruct shapes in a creative manner (e.g., represent the semicircle leg with bars).

diverse captions than 3DEncoder-LSTM (Vinyals et al., 2015; Chen et al., 2021) due to the help of involving Gumbel noise over the learned priors in the inference. This is also shown in Figure 7 where Shape Compiler can describe given shapes in multi-angles. Also, our framework achieves generally higher BLEU, CIDER, and ROUGE scores than 3DEncoder-LSTM. By inspecting the predictions, we found that 3DEncoder-LSTM tends to predict the frequent words in datasets (e.g., back and legs), and the output descriptions are less meaningful in semantics. However, those repetitive words can help them achieve higher BLEU, CIDER and ROUGE scores in Text2Shape and ShapeGlot. Therefore, evaluating *Point Cloud ⟹ Text* performance with both quality and diversity metrics is essential. Furthermore, the ground-truth descriptions differ significantly from our predictions on diversity metrics, suggesting that the proposed method still has huge gaps in achieving human-level shape captioning.

### 4.3 *Point Cloud ⟹ Program*

Shape Compiler performs transformation: *Point Clouds ⟹ Program*, which can help us understand how the point clouds are assembled by parts and regularities. For example, in Figure 8, we can tell how to decompose the point clouds into primitives via the symbolic words in the program and also find the mapping between points and the corresponding primitive via executing the program. Furthermore, the symmetry relationship between those racks of the cabinet is implicitly encoded in the *FOR* statements. The discovery of parts and relationships can help us design robots better interacting with 3D objects.

Neural Program Generator (Tian et al., 2019) is our most direct baseline. They used a two-layered LSTM to gradually generate program blocks and programs inside each block to form the final shape of programs. We will not adopt the guided adaptation used in (Tian et al., 2019) in our experiments because it needs extra training loops and prevents the practical use of program generation (further discussion is included in Appendix D). We also compare with CSGNet (Sharma et al., 2018), which applies boolean operations on shape primitives and assemble shape recursively.

| Metric | Method | Chair | Table | Bed | Sofa | Cabinet | Bench | Avg |
|---|---|---|---|---|---|---|---|---|
| IoU ↑ | CSGNet (Sharma et al., 2018) | 0.365 | 0.406 | - | - | - | - | - |
| | Program Generator (Tian et al., 2019) | 0.438 | 0.517 | 0.254 | 0.324 | 0.304 | 0.216 | 0.342 |
| | Shape Compiler Limited | 0.429 | 0.539 | **0.27** | 0.31 | 0.504 | 0.314 | 0.394 |
| | Shape Compiler | **0.492** | **0.634** | 0.252 | **0.432** | **0.51** | **0.451** | **0.462** |
| CD ↓ | CSGNet (Sharma et al., 2018) | 7.73 | 7.24 | - | - | - | - | - |
| | Program Generator (Tian et al., 2019) | 1.64 | 1.97 | 4.78 | 3.14 | 2.95 | 2.71 | 2.87 |
| | Shape Compiler Limited | 1.53 | 1.21 | 4.51 | 2.84 | 1.58 | 1.57 | 2.21 |
| | Shape Compiler | **1.17** | **0.83** | **4.39** | **2.52** | **1.49** | **1.38** | **1.96** |

Table 4: *Point Cloud ⟹ Program.* CD is multiplied by $10^2$. Avg denotes average number among categories.

**Evaluation Protocol:** Shape Program (Tian et al., 2019), CSGNet (Sharma et al., 2018), and our models will predict programs under specific domain languages. By executing the output programs, we can obtain volumetric representation shapes; then, we can compute IoU based on the output voxels. Also, we sample points over the surface of output voxels for computing Chamfer distance. Due to the benefits of being a probabilistic method, our framework can predict many programs given a single point cloud. We here generate 48 programs for comparisons, the same number we used in our text tasks. Ablation studies of the generated program numbers are included in Appendix D.

**Result:** According to Table 4, Shape Compiler outperforms all the methods due to the benefits of multimodal learning on all tasks and data. It also suggests our framework can effectively take advantage of task-unrelated heterogeneous data. Figure 8 shows two positive examples of our framework that better handles novel data than the baseline. Both cases show that the compared baseline mainly relies on memorizing the training table data, while Shape Compiler successfully predicts the rough structure and the regularities. Regarding the scores, Shape Compiler Limited achieved comparable performance with Shape Generator, while the significant performance gap in cabinet and bench categories shows our framework has stronger generalizability. Due to the limitation of the current program grammar, our method cannot handle complex structures for now. We raise attention to the next-level shape program, and some further discussion is included in Appendix D.

### 4.4 *Partial Point Cloud $\implies$ Complete Point Cloud*

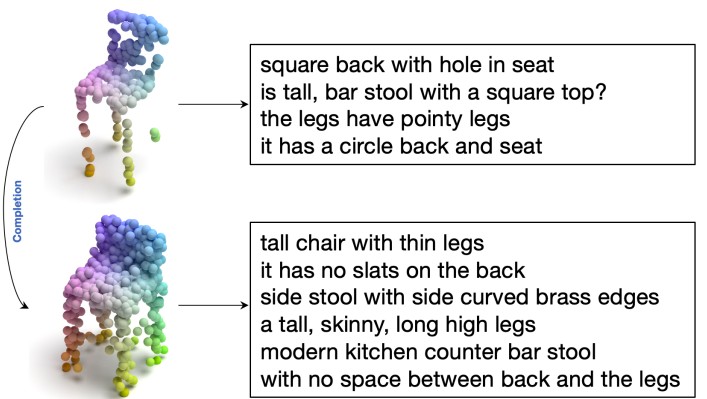

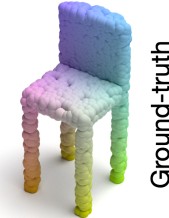

| Method | Chair | Airplane | Car |
|---|---|---|---|
| PointFlow (Yang et al., 2019) | 6.93 | **1.07** | 3.97 |
| PVD (Zhou et al., 2021) | 7.34 | 1.19 | 3.83 |
| Shape Compiler Limited | 7.57 | 1.62 | 4.82 |
| Shape Compiler | **6.9** | 1.56 | **3.81** |

Figure 9: The success of shape completion boosts shape structural understandings. We use Neural Shape Compiler to do *Point Cloud $\implies$ Text* for both the partial chair and the complete chair. The ground-truth shape is on the top right.

Table 5: *Partial Point Cloud $\implies$ Complete Point Cloud.* Numbers are Chamfer distance ↓ with multiplying $10^2$.

In this section, we extend Neural Shape Compiler to conduct *Partial Point Cloud $\implies$ Complete Point Cloud* (shape completion) task by projecting both partial and full point clouds into discrete codes and performing conditional generation thereon. We followed the benchmark provided in (Zhang et al., 2018; Zhou et al., 2021) and the detailed data preparation is listed in Appendix A.3. Different from the way used in Zhou et al. (2021), we train models on all training shapes in a class-agnostic way. Results in Table 5 show Shape Compiler achieved great performance compared with the previous method. Also, PVD (Zhou et al., 2021) is slower than Shape Compiler because they involve very long steps (1,000) in the diffusion process.

Additionally, with the help of Neural Shape Compiler, we may better share the progress across different tasks. In Figure 9, we individually perform *Point Cloud $\implies$ Text* over the partial chair and the complete chair. The ground-truth chair is on the right. For the partial chair, the captioning results include hallucinated descriptions, like *hole in the back* and *circle back*, and also accurate descriptions of the partial chair but not precisely related to the ground-truth one, like *pointy legs.* Compared to it, the descriptions for the completion result are more related to our ground-truth point cloud.

# 5 Limitations and Future Works

This paper proposed Neural Shape Compiler, a unified framework to perform multimodal inference and achieved great performance on *Text* $\implies$ *Point Cloud*, *Point Cloud* $\implies$ *Text*, *Point Cloud* $\implies$ *Program*, and Point Cloud Completion tasks. However, there are still a lot of things to be considered:

1. Similar to (Chen et al., 2018; Liu et al., 2022b; Mittal et al., 2022), our method requires large-scale paired data, where the 3D-text pairs are hard to obtain. This situation could be alleviated by the development of large-scale 3D-text datasets and recovery of 3D data from 2D images (Gkioxari et al., 2022) and videos (Qian et al., 2022) that contain paired text information.

2. Our framework is extensible, where we showed a case in shape completion task (Table 5). Similarly, we can project images (Van Den Oord et al., 2017), tactile information (Gao et al., 2022), and other modalities into the discrete representation and learn over them via a unified framework for various downstream tasks, such as single image 3D reconstruction (Zhang et al., 2018), multiview reconstruction (Ji et al., 2017) and multi-modal grasping (Calandra et al., 2018). On the other hand, it would be interesting to study extending diffusion-based frameworks to handle various types of generative modeling, since the current diffusion models usually focus on one type of transformation Rombach et al. (2022); Gong et al. (2022).

3. The proposed framework currently cannot well handle out-of-distribution geometry, such as the dragon in (Morreale et al., 2022) and the Hilbert cube in (Liu et al., 2020a). Figure 10 shows the results of *Point Cloud* $\implies$ *Text* and *Point Cloud* $\implies$ *Program* over the Hilbert cube by Shape Compiler. The current programs can only tell the coarse geometry, and captions are about similar objects in our training data, such as cubes and cabinets.

Figure 10: *Point Cloud* $\implies$ *Text* and *Point Cloud* $\implies$ *Program* results over the input Hilbert cube point clouds by Shape Compiler. One description per sentence.

4. Our approach currently adopts the domain-specific language (DSL) proposed in (Tian et al., 2019) as one of the input and target shape abstractions. However, the DSL's grammar and syntax are not powerful enough to handle various complex shapes. For example, it fails to perform accurate reconstruction or completion for 3D shapes with geometrical details (Chaudhuri et al., 2020) and detect self-symmetries possessed by individual parts (Mitra et al., 2006). Since the progress of shape programs is orthogonal to the proposed framework, the development of shape programs can further benefit our framework. Therefore, this paper also raises attention to developing the next level of shape programs.

5. Neural Shape Compiler still generates outputs that are mismatched to the input conditions. For example, it may generate shapes unsatisfied with the input text in *Text* $\implies$ *Point Cloud* task and captions wrongly describe input point clouds in *Point Cloud* $\implies$ *Text* task.

6. Our current framework and data focus on correctly modeling connections between 3D geometry and text. However, less attention has been paid to color, texture, and material, which will be investigated in the future with the support of appropriate data.

7. DALL-E (Ramesh et al., 2021) ranks their outputs of (*Text*, *Image*) based on the CLIP embedding similarity scores (Radford et al., 2021). However, as the experiments conducted in Section 4.1, the CLIP similarity score cannot properly reflect the alignment between 3D shape and text if the text contains geometric details. How to align 3D objects and text is a fundamental problem we should study in the future.

8. Neural Shape Compiler currently compiles different shape abstractions. Compared to this, a further direction is to compile operations on one shape abstraction into practical steps in other abstractions and change it accordingly. For example, we add "*armless*" into a text description of a chair with two arms. This change can be effectively compiled into operations "*remove two arm parts and their symmetry relationship*" in

the shape hierarchy and "*delete all points belonging to the two arms*" in point clouds. We leave the shape editing direction to future study.

## 6 Conclusion

We proposed Neural Shape Compiler to transform between three shape abstractions: *Text*, *Point Cloud*, and *Program*. With the help of *Point*VQVAE, it achieved great performance in *Text $\Longrightarrow$ Point Cloud*, *Point Cloud $\Longrightarrow$ Text*, *Point Cloud $\Longrightarrow$ Program*, and PointCloud Completion tasks via a unified and extendable framework. Our experiments show that Neural Shape Compiler benefits from joint training on all heterogeneous data and tasks. Despite showing promises, Shape Compiler has limitations and draws attention to several related directions for future research (Section 5). Furthermore, we study CLIP embeddings in text-guided shape generation and find that it is unsuitable for use once the text contains geometric details. We hope that Neural Shape Compiler can serve as an effective framework for connecting different shape abstractions, and the studies in this paper can facilitate the progress in 3D multimodal learning research.

## 7 Acknowledgement

This work is supported by a fellowship from UMich Rackham Graduate School, two grants from LG AI Research, and Grant No. 1453651 from NSF. We thank Mohamed El Banani and Janpreet Singh for helpful discussions on text-to-3D and the help in proposing the CLIP text similarity experiment. We thank Mohamed EI Banani for his helpful comments on our writing. We thank Chris Rockwell for the helpful discussion at the early stage of this project, especially in *Point*VQVAE. We thank Joyce Zhang and Yutong Tan for parts of the aesthetic design and suggestions. We thank Kaiyi Li for the helpful discussion on compiler concepts and for helping us compile several renderers in Apple-M1-Max chip. Tiange especially thanks Minghua Liu, Zhiwei Jia, Weijian Xu, Xiaoshuai Zhang, Yuheng Zhi, Zhankui He, Zexue He, Kaichun Mo, Eric Yi, and Jiajun Wu for helpful conversations back in 2020.

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

# Appendix

## Table of Contents

## A  Data Collection

### A.1  3D Shape Assets

As mentioned in Section 4, we collect a total of 140,419 shapes from ShapeNet (Chang et al., 2015), ABO dataset (Collins et al., 2021), and Shape Program (Tian et al., 2019). For ShapeNet, we use the most recent release version (ShapeNetCore.v2), which contains 52,472 shapes from 55 categories. ABO dataset contains 7,947 shapes from 98 categories and has nine overlapped shape categories with ShapeNet. We adopt the 4 Chair and 10 Table templates in Shape Program (Tian et al., 2019) to synthesize 40,000 shapes in each category. We do not distinguish shape categories during training for both *Point*VQVAE and ShapeCode Transformer. We will normalize the input point clouds into a unit ball for addressing dataset gaps.

### A.2  Shape-Text Pairs

We collect 107,371 (*Point Cloud, Structure-Related Text*) pairs by adjusting and annotate the current datasets, including Text2Shape (Chen et al., 2018), ShapeGlot (Achlioptas et al., 2019), and ABO (Collins et al., 2021) datasets. We take tables from Text2Shape, chairs from ShapeGlot, and all shapes from the ABO dataset. Therefore, our dataset consists of furniture, most of which are tables and chairs. We list our considerations and data collection details below.

**Text2Shape:** (Chen et al., 2018) provides descriptions for both chairs and tables. Through careful inspection, we found there are a large number of descriptions regarding artificial colors (e.g., red, blue), textures (e.g., green grids on the top), and materials (e.g., wooden, steel). Since this work aims to focus on the 3D geometry and will not predict any colors for output point clouds, we delete those artificial texture contents but remain the geometrical ones. Also, compared to Text2Shape, ShapeGlot (Achlioptas et al., 2019) provides much more geometrical descriptions for chairs, where the chairs have a very high overlap with chairs in Text2Shape. We thus discard the chair descriptions and only adopt the table descriptions of Text2Shape in our data. For a few data which lack structural descriptions, we will annotate them manually. Some examples are in Figure 11.

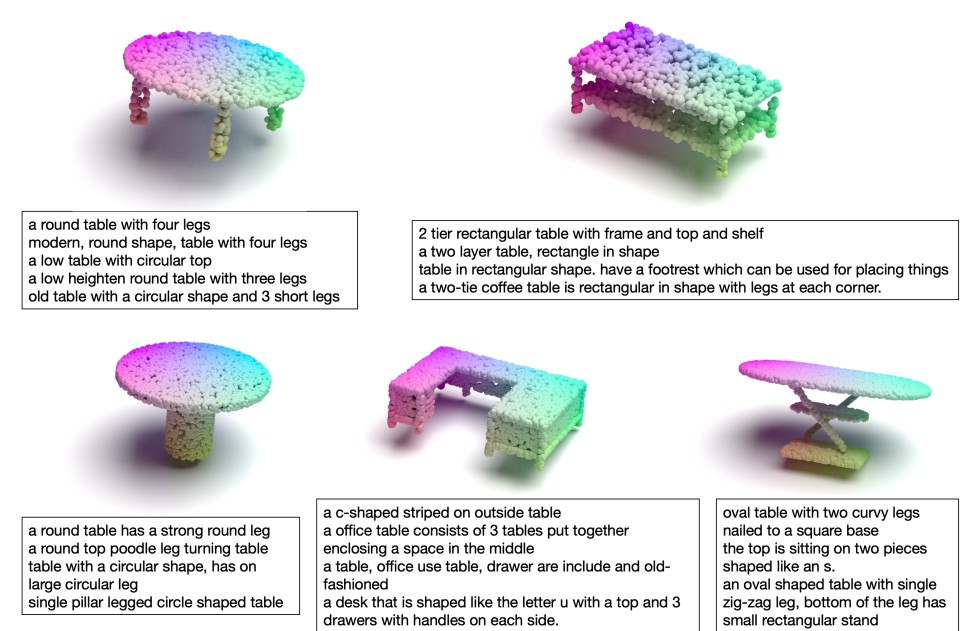

Figure 11: Random examples in adjusted Text2Shape dataset. One description per sentence.

**ShapeGlot:** (Achlioptas et al., 2019) provides many meaningful structural descriptions for chairs. However, the descriptions are about triple relationships. It provided users with two shapes and a comparative description and asked users to choose which shape most satisfied the description, i.e. (*Shape A*, *Shape B*, *Text*). We transfer the triples into doubles by assigning the text to the target shape according to annotations. However, some texts lose their meanings if they do not have the control shape. Furthermore, we found single text in ShapeGlot is usually not very informative that can plot the target shape accurately. In the end, we filter out the less-meaningful texts and concatenate several shapes' texts together as one description. Also, we remove superlative words, like longest, because they usually cannot hold across the entire dataset. For a few data which lack structural descriptions, we will annotate them manually. Some examples are in Figure 12.

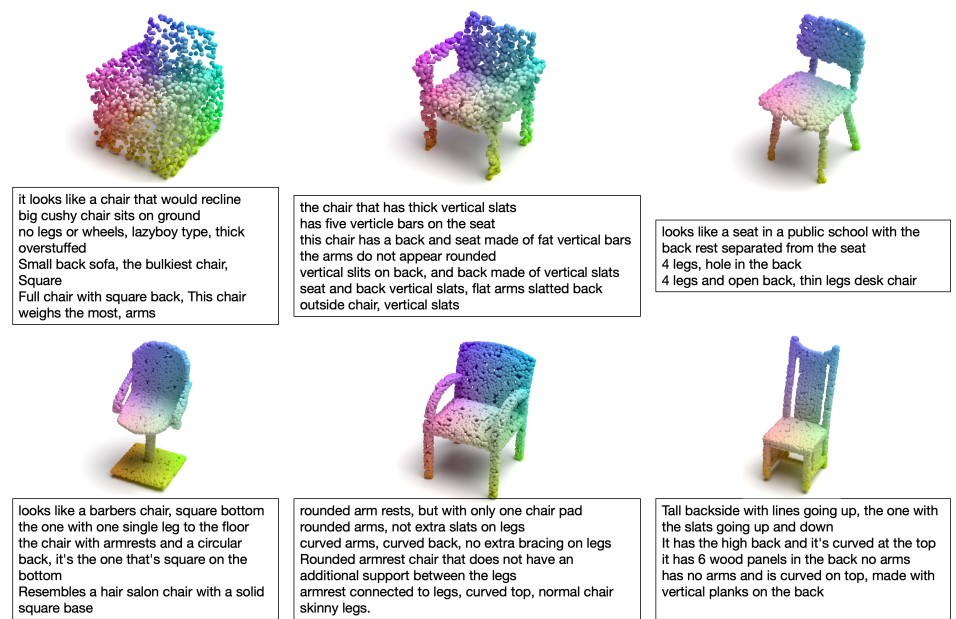

Figure 12: Random examples in adjusted ShapeGlot dataset. One description per sentence.

**ABO:** (Collins et al., 2021) provides high-quality CAD models and descriptions in different languages for amazon products. We only use the description of the 'item_name' under the language tag 'en_US'. To add structure-related texts to ABO shapes, we collect more descriptions by manually annotating. Also, since ABO consists of amazon products, some original descriptions contain brand names, such as "amazon basics" and "ravenna", which are not connected to shape geometry. Also, some descriptions contain product dimensions like $16.0 \times 8.2 \times 7.4$. Since we normalize all point clouds into a unit ball, those dimensions are not connected to its scale. Therefore, we deleted the descriptions related to brand names and dimensions in the descriptions. Some examples are in Figure 13.

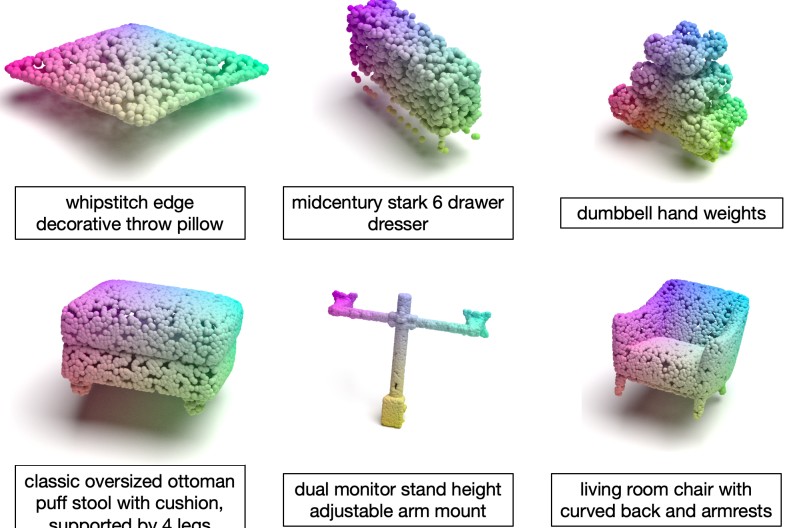

Figure 13: Random examples in adjusted ABO dataset. One description per sentence.

### A.3 ParitalShape-CompleteShape Pairs

We used the data from GenRe (Zhang et al., 2018) which has 20 random view renderings of airplanes, cars, and chairs from ShapeNet, for a total of $20 \times 9{,}646$ training (*point-clouds*, *point-clouds*) pairs and $20 \times 1{,}382$ test (*point-clouds*, *point-clouds*) pairs. We sample 200 points from the above depth images as inputs, sample 2048 points from the corresponding 3D shapes as targets, and evaluate over all the provided 20 views. Unlike previous benchmarks (Zhou et al., 2021), which train a model for a single shape category, we train every completion model with all training data.

## B Training & Model Details

This section provides the training and model details for our *Point*VQVAE and ShapeCode Transformer used in our main paper, and the *Program*VQVAE we mentioned in Section 3.1. We implement and train our models with PyTorch (Paszke et al., 2019) and 8 A40 GPUs.

### B.1 *Point*VQVAE

For *Point*VQVAE, we adopt the cosine annealing strategy of SGDR (Loshchilov & Hutter, 2016) (w/o restarts) as our optimization scheduler and set the initial learning rate 0.002 with $8 \times 32$ batch size and 200 epochs.

*Point*VQVAE encoder has two hierarchical layers ($\#T = 2$). In the first layer, we sample 512 center points. Then, we gather information from 64 points within $R_{T_1} = 0.1$ ball around each center point with two fully connected layers [64, 512] and [512, 512]. After the first layer, our point cloud size will be reduced to 512 from 10,000 (original input size). Similarly, in the second layer, we sample 128 center points, and do $R_{T_2} = 0.4$

ball query with 512 points. We also adopt two fully connected layers here with shapes [512, 512]. After the encoding phase, we will have 128 embeddings and look for the closest embedding in the codebook that has the size of [512, 512]. We concatenated the indexed embedding to form a [128, 512] embedding and sent it to the decoder. The structure of *Point*VQVAE decoder is shown in Figure 2, we have batch norm 1D (Ioffe & Szegedy, 2015) and ReLU (He et al., 2015) following each convolutional layer. The output of the residual-like structure (Figure 2) has the shape [#Batch Size, 512, 128]. We then applied a max-pool layer to the last dim of the output to obtain an embedding with shape [#Batch Size, 512, 1]. The max-pool layer can eliminate the effect caused by the concatenation order of the codebook embedding, similar to the use of max-pool layers in PointNet (Qi et al., 2017a). After the max-pool layer, we finally have two Convolutional 1D layers with [512, 1024] and [1024, 2048 * 3] to predict 2,048 points. Batch norm 1D and ReLU are also used between the last two layers.

We use Chamfer Distance (CD) and Earth Mover's Distance (EMD) as the reconstruction loss to serve as one of the objectives of *Point*VQVAE. Specifically, we adopt the implementation of CD in Pytorch3D (Ravi et al., 2020) and implemented an EMD approximation via auction algorithm (Bertsekas & Castanon, 1989) with the help of codes provided in (Fan et al., 2017; Liu et al., 2020b). The EMD codes will be released with a detailed report.

$$\mathcal{D}_{EMD}(P_1, P_2) = \min_{\phi:P_1 \to P_2} \sum_{x \in P_1} \|x - \phi(x)\|_2, \quad \phi \text{ is a bijection} \tag{2}$$

$$\mathcal{D}_{CD}(P_1, P_2) = \sum_{x \in P_1} \min_{y \in P_2} \|x - y\|_2^2 + \sum_{y \in P_2} \min_{x \in P_1} \|x - y\|_2^2 \tag{3}$$

$$\tag{4}$$

The overall objective for optimizing *Point*VQVAE adopted the one used in (Van Den Oord et al., 2017) and is shown below, where sg represents stop gradient operations, $\mathcal{Z}_e(p)$ refers to the output of our encoder, $e$ is the nearest embedding in our codebook with the encoder output. The first two terms are the reconstruction loss to push the final output of *Point*VQVAE closer to input point clouds. The third term is $\mathcal{L}_2$ loss to move the embedding in our codebook closer to the corresponding encoder output. The fourth one is the commitment loss to constrain those embedding growing.

$$\mathcal{L}_{total} = \mathcal{D}_{EMD} + \mathcal{D}_{CD} + \|\text{sg}[\mathcal{Z}_e(p)] - e\|_2^2 + \|\mathcal{Z}_e(p) - \text{sg}[e]\|_2^2$$

## B.2 ShapeCode Transformer

We also adopt the cosine annealing strategy as our optimization scheduler and set the initial learning rate 0.001 with $8 \times 24$ batch size and 100 epochs. For generality, we adopt the simplest Transformer (Vaswani et al., 2017) with depth 5, wide 64, and 8 attention heads. We integrate full attention masks (Child et al., 2019) to force it to do autoregressively predictions (Van den Oord et al., 2016).

As mentioned in Section 3.2, we have different positional embedding for each type of data pair (i.e., task), which will be optimized during the training. Since we pad 0 at the leftmost of our data, the positional embedding for the conditional code will have an extra token. For example, if we conduct (*Text*, *Point Cloud*) transformation, we will have $P_{text} = 256 + 1 = 257$ positional tokens for Text and $P_{point} = 128$ for Point Cloud. Therefore, we have a total of $257 + 128 = 385$ positional tokens for (*Text*, *Point Cloud*). For each positional embedding, we have the embedding dimensional 512. As a result, we have embedding size $[385, 512]$ for the task (*Text*, *Point Cloud*). For each task, Neural Shape Compiler has different positional embeddings but the same process described above.

Also, ShapeCode Transformer has a self-maintained embedding matrix for each type of code, as illustrated in Section 3.2. We also use (*Text*, *Point Cloud*) transformation as example, where Text code has $N_{text} = 256$ tokens and Point Cloud code has $N_{point} = 128$. The embedding dimension is 512. Therefore, we have embedding matrix with size $[N_{text}, 512]$, $[N_{point}, 512]$, and $[N_{program}, 512]$ for Text, Point Cloud, and Program, respectively. Regarding token length and vocabulary size for each shape abstraction, $\mathcal{C}^{point}$ has

$N_{point} = 128$ tokens with a vocabulary size of 512 via argmaxing from the $Point$VQVAE encoder output; $C^{text}$ has $N_{text} = 256$ tokens of vocabulary size $49,408$, we pad between the last valid text token and the beginning of another token with a value between $[49408, 49408 + 256]$; $C^{program}$ has $N_{program} = 240$ tokens and a vocabulary size of 78.

The overall objective of ShapeCode Transformer is the summation of loss computing over each pair of data: (1) (*Text, Point Cloud*); (2) (*Point Cloud, Text*); (3) (*Point Cloud, ShapeProgram*); (4) (*Partial PointCloud, Complete PointCloud*). For each pair, we define the loss over every token with Cross-Entropy loss as shown below, where $\mathcal{L}$ indicates cross-entropy. Noting that, for program codes $C^{program}$, we will turn negative integers into positive integers, like mapping [-20, 20] to [0, 40].

$$\sum_k \mathcal{L}(c_k, c_k^{gt}|[0, \cdots, c_{k-1}])$$

### B.3 *Program*VQVAE

Currently, Shape Programs (Tian et al., 2019) only has limited discrete parameters, and thus, we adopted the way of discretizing parameters into codes in our main paper. For example, we will map integers in [-20, 20] to [0, 40] to assign unique positive integers for each parameter. Although the current shape program can already cover a lot of 3D shapes, it may be upgraded to encode more detailed structures of 3D shapes that need to involve continuous parameters. For example, the statement *draw('Top', 'Square', P=(-1,0,0), G=(2,5))* in Figure 1 may have decimal parameters *draw('Top', 'Square', P=(-1.2,0,0.3), G=(2,5.7))*. Furthermore, the primitive basis used in the shape program may be replaced with a functional basis (Mescheder et al., 2019; Genova et al., 2020). As mentioned in Section 3.1, we also designed *Program*VQVAE to transform shape programs into codes to fill the gap in handling continuous parameters.

The design of *Program*VQVAE is pretty similar to our *Point*VQVAE, where we leverage the most basic concepts of vector quantization and adopt the straight-through gradient estimator and the vanilla codebook objective used in (Van Den Oord et al., 2017) for optimizing. Given a statement, we will turn its statement type into a one-hot vector and concatenate it with its parameters as the input to *Program*VQVAE. For example, we will turn *draw('Top', 'Square', P=(-1,0,0), G=(2,5))* into $[3, -1, 0, 0, 2, 5, 0, 0]$ where the value 3 is the represents the statement type 'draw'. The encoder and decoder will be two vanilla Transformer (Vaswani et al., 2017) with depth 3 and MLP dimension 32, and the codebook is [512, 512], the same size as we used in *Point*VQVAE. We use cross-entropy to compute the loss for predicting statement types and $\mathcal{L}_2$ loss for regressing statement parameters.

We synthesize 200,000 (*Point Cloud, Program*) pairs for training the *Program*VQVAE. On 10,000 validation data, it achieved 100% accuracy in predicting statement types and $< 0.01$ L2 distance in parameter regression. The results show the proposed *Program*VQVAE can well learn the distribution of our synthesized shape programs.

### B.4 Baselines Implementation Details

This paper compared various baselines across different tasks to examine the performance of the proposed framework. We generally follow their provided codes and default configurations for each baseline, including training epochs, learning rates, and other hyper-parameters. We only modify the code when changes are needed to use our dataset. Some modifications are listed in this section.

- Shape IMLE (Liu et al., 2022b) preprocesses all texts with Bert (Devlin et al., 2018) in its framework. We adopted the same way and used *BertTokenizer* of the pre-trained flag "bert-base-uncased" to process all the texts in our dataset.

- We adopted *clip.tokenize* and *clip.encode_text* to encode all test texts and input them into CLIP-Forge (Sanghi et al., 2022) for quantitative comparisons. We input the text prompt to DreamField (Jain et al., 2021) by the provided command line in Gihub of Jain et al. (2021) for qualitative comparisons. In our environment, we need 4 GPUs of 2048 Gb memory to run a single text input with DreamField

(Jain et al., 2021). It takes more than eight days to finish the default 10,000 iterations for single text input. Therefore, it is impractical to test DreamField in our test set, which consists of 15,000 text-shape pairs.

- CWGAN (Chen et al., 2018) provided processed data as templates. We follow their data formats to process our data and update each entry, including *caption_tuples*, *caption_matches*, *vocab_size*, *max_caption_length*, and *dataset_size*. We adopted spaCy (Honnibal et al., 2020) to tokenize texts as mentioned in Chen et al. (2018). (Chen et al., 2018) simply set the token number in a linear increase rule based on their data order, while we set our token number based on our data order. We also disregarded descriptions with more than 96 tokens as (Chen et al., 2018) did.

- For AutoSDF (Mittal et al., 2022), their current codebase did not support training generation models conditioned in texts; we do not compare it for now. We will try to add their results once the code is updated.

- For the point cloud completion baseline PVD (Zhou et al., 2021), we jointly train their models with three data classes. This differs from their original paper, which trains separate models for each shape class.

## C  *Point*VQVAE Discussions

### C.1  Analysis of Learned Codebook

This section provides some preliminary analysis of the learned codebook of *Point*VQVAE. The codebook is the interface to connect the output of *Point*VQVAE encoder and the input of *Point*VQVAE decoder. In our experiments, the codebook size is [512, 512], and we have 128 embedding for a single input point cloud.

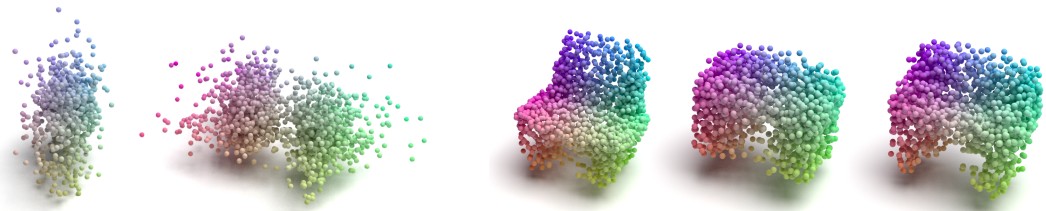

Figure 14: The right three shapes are reconstruction results of random codes and the left two shapes are reconstruction results of indexed codes illustrated in the text.

For each input point cloud, we will have 128 codes to index the embedding and feed it into the decoder later. We look at how many different embeddings of the codebook will be used for a single input point cloud. We compute the number with whole ShapeNet (Chang et al., 2015) shapes, where we auto-encode shape and compute the average ratio of unique codes. The result is 44.28, which indicates point cloud code $\mathcal{C}^{point}$ will have multiple replicate code numbers. Does this indicate that the codebook learns the basic primitives (Deprelle et al., 2019), selects the suitable ones for a specific point cloud, then fuses them for reconstruction? If so, will those primitives be perceptually understandable to our humans? To study this, we design indexed codes with all one number $[x, x, \cdots, x]$, like $[1, 1, \cdots, 1]$, length = 128. Those codes visualize the codebook's $\#x$ embedding. We feed the special codes into our decoder to reconstruct the corresponding embedding and show two examples at the left of Figure 14. Results show the embedding is pretty random to humans and may exist in certain symmetry structures. Since the reconstruction results are highly correlated to the decoder weights, the way we adopted above may not be the ideal solution. We will try to develop other methods to understand how the 3D codebook works in the future. Besides, we also tried to randomly generate the codes and reconstruct them to see if the random samples from the latent discrete space are meaningful. Results are shown in the right of Figure 14, where they are somewhat reasonable in the structure. We also show an example in Figure 15 where we interpolate two different shape codes and reconstruct the intermediate results.

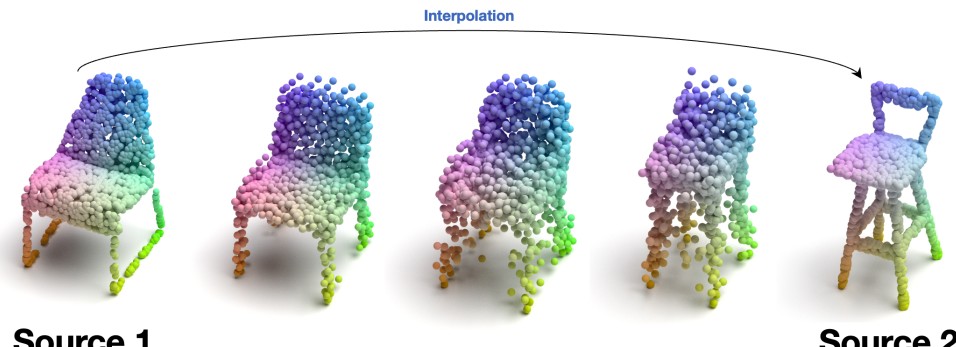

Figure 15: A random example of interpolating latent codes between two resource shapes of our proposed *Point*VQVAE.

## C.2 Empirical Studies

In this section, we conduct empirical studies about *Point*VQVAE, including ablation studies and comparisons. We changed the number of points in the last layer of the encoder (the color points in Figure 3) from 1 to 128. When the point number equals 1, our encoder equals the standard encoder of PointNet++ (Qi et al., 2017b) which uses a global pooling layer. Table 6 left shows such a design brings a catastrophic performance drop due to it being significantly difficult for the codebook to encode holistic point clouds instead of parts of the point clouds.

| Variants | CD ↓ |
|---|---|
| *Point*VQVAE globalpool | 16.63 |
| *Point*VQVAE final-16 | 10.26 |
| *Point*VQVAE final-32 | 8.57 |
| *Point*VQVAE final-64 | 7.34 |
| *Point*VQVAE final-128 | 6.93 |
| *Point*VQVAE final-256 | 6.78 |

| Variants | CD ↓ |
|---|---|
| $[R_{T_1} = 0.1, R_{T_2} = 0.2]$ | 13.356 |
| $[R_{T_1} = 0.1, R_{T_2} = 0.4]$ | 6.877 |
| $[R_{T_1} = 0.1, R_{T_2} = 0.6]$ | 8.283 |
| $[R_{T_1} = 0.05, R_{T_2} = 0.1, R_{T_3} = 0.6]$ | 6.513 |
| $[R_{T_1} = 0.05, R_{T_2} = 0.2, R_{T_3} = 0.6]$ | 7.808 |

Table 6: Ablation studies over *Point*VQVAE. CD is multiplied by $10^3$. **Left**: X in *Point*VQVAE final-X means how many points we sampled in the last layer of the encoder. "globalpool" indicates only having one point with the global receptive field. **Right**: studies the choices of $R_{T_i}$ and the number of downsampling processes in *Point*VQVAE (Section 3.1).

We also perform some ablation studies over the number of downsampling processes and choices of $R_{T_i}$ in Section 3.1. We've tested several ablations about $R_{T_i}$ as shown in the right of Table 6, including both two and three times downsampling processes. In our experiments, we chose $[R_{T_0} = 0.1, R_{T_1} = 0.4]$ due to its good performance and light weight.

Besides, we compare with some point clouds based auto-encoders (Achlioptas et al., 2018; Yang et al., 2019; Luo & Hu, 2021; Cheng et al., 2022). For each shape, we sample 2048 points thereon and use Chamfer Distance (CD) to measure the difference between inputs and reconstruction results. We will report Chamfer Distance $\mathcal{D}_{CD}(P_1, P_2)$ as the summation of both two-direction $P_1 -> P_2$ and $P_2 -> P_1$. EMD is not reported, because each method adopted the EMD implementation proposed in (Fan et al., 2017), which cannot guarantee an approximation of Earth mover's distance. In contrast, all the used CD implementations are correct. The results are shown in Table 6. The numbers under "Oracle" mean the lower bound of reconstruction, obtained by computing the distance between two different samplings from the same shape meshes. Compared to *Point*VQVAE Limited, *Point*VQVAE achieved

| Methods | CD ↓ |
|---|---|
| l-GAN (Achlioptas et al., 2018) | 9.23 |
| Point Flow (Yang et al., 2019) | 7.76 |
| ShapeGF (Cai et al., 2020) | 5.97 |
| DiffGen (Luo & Hu, 2021) | 5.62 |
| *Point*VQVAE Limited | 6.93 |
| *Point*VQVAE | 5.98 |
| Oracle | 3.09 |

Table 7: Autoencoding performance. CD is multiplied by $10^3$.

higher scores which mean training over more 3D data can help *Point*VQVAE model the 3D shape distribution and also not reduce its performance on the part of the training data.

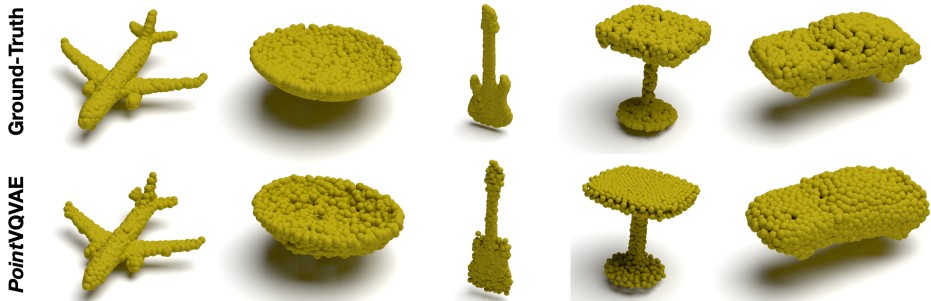

Figure 16: Random reconstruction examples of *Point*VQVAE.

## D *Point Cloud ⟹ Program* Discussions

In this section, we mainly discuss two points around *Point Cloud ⟹ Program*: (1) the generated sample numbers used in our shape program experiments; (2) the guided adaptation used in (Tian et al., 2019).

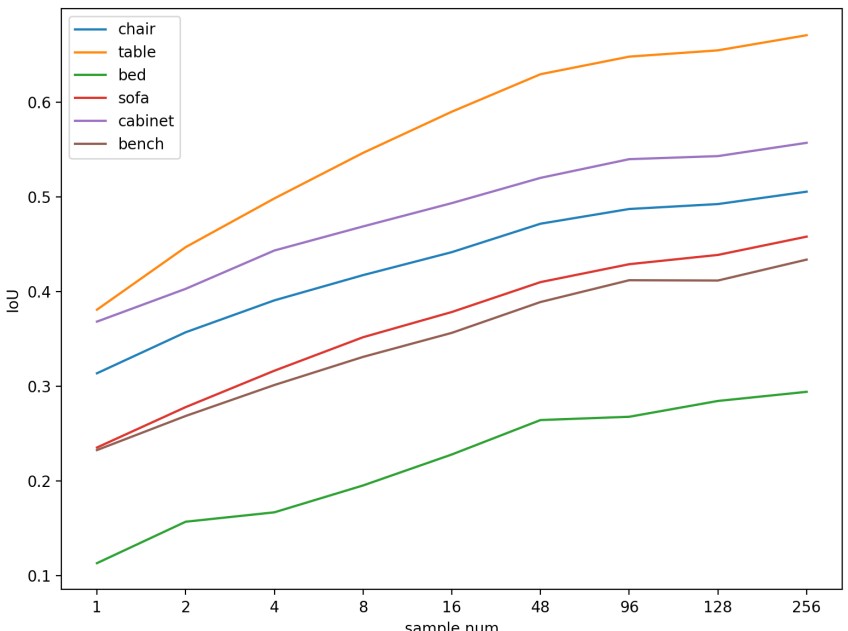

Figure 17: *Point Cloud ⟹ Program* performance versus the shape generated sample numbers.

We adopt 48 as the sample number in *Point Cloud ⟹ Program* experiments because all other tasks adopt batch size 48. Here, we provide a more thorough study of the sample numbers. We test Shape Compiler performance in all six shape categories with sample numbers [1, 2, 4, 8, 16, 48, 96, 128, 256]. The results are shown in Figure 17. The results show that the proposed method achieved higher performance by increasing the generated sample numbers. This is also one of the benefits of the generation framework we adopted in our approach.

Shape Program (Tian et al., 2019) used guided adaptation in finetuning their models to new test data. They achieved higher performance by using guided adaptation. However, in our experiments, we intended not to follow (Tian et al., 2019) to perform guided adaptation. We choose this way because we aim to

take a step further in shaping program generation for practical uses. In real-world applications, we usually cannot afford the extra training time for conducting "guided adaptation"; sometimes, we even do not have enough computation to run adaptation (e.g., edge computing). Also, generating shape programs without "guided adaptation" can enable us to use one class-agnostic model to process shapes from different categories instead of having class-specific models for each class. The proposed framework, Shape Compiler, only has one class-agnostic model for processing six different shape categories. However, Tian et al. (2019) will need to train six different models to process the test shape if they use guided adaptation. Combining the advantages of time-saving and class-agnostic, PointCloud-to-Program could take a step toward achieving real-world impact.

# E   More Results

In this section, we provide more generated samples by the proposed Neural Shape Compiler. Extra results are contained in the supplementary.

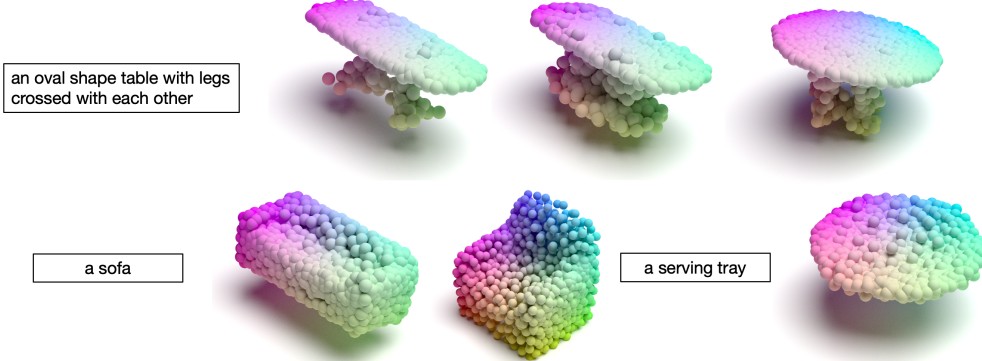

Figure 18: Generated shapes by Neural Shape Compiler given different text prompts. Shape Compiler can generate multiple shapes given a text query.

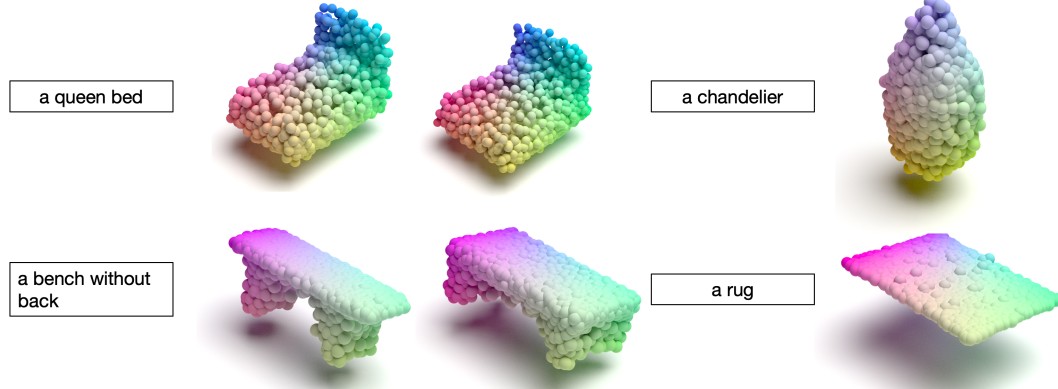

Figure 19: Generated shapes by Neural Shape Compiler given different text prompts. Shape Compiler can generate multiple shapes given a text query.

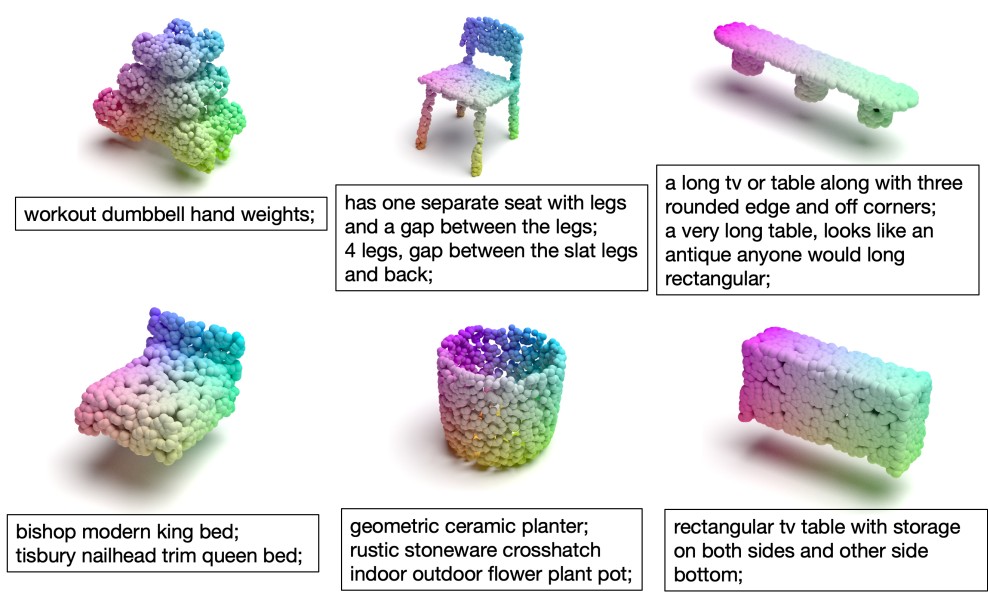

Figure 20: *Point Cloud ⟹ Text* results by Neural Shape Compiler given different point clouds. Shape Compiler can generate multiple captions given a point cloud query. One description per sentence.

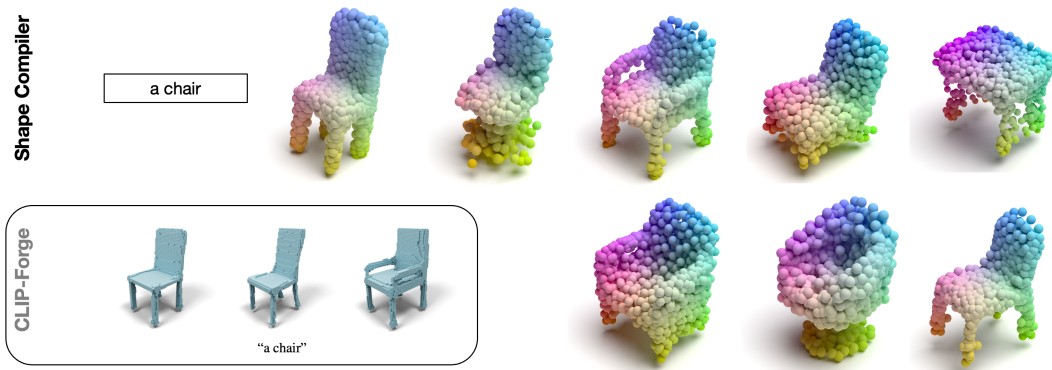

Figure 21: Directly comparing the performance of Neural Shape Compiler with CLIP-Forge (Sanghi et al., 2022) results on a simple text prompt.

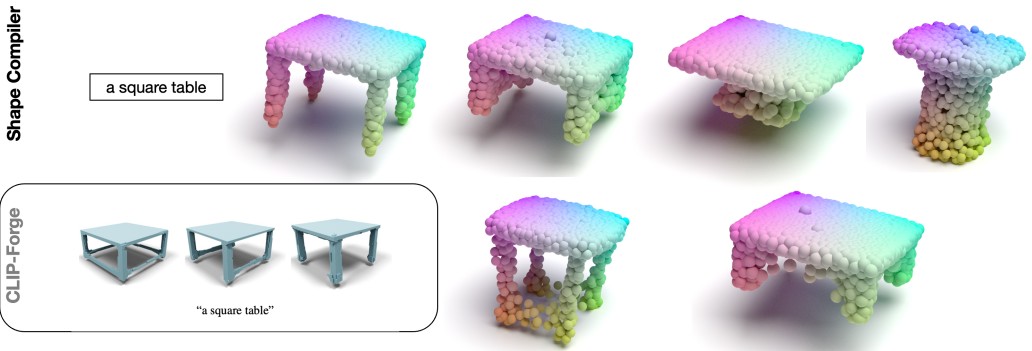

Figure 22: Directly comparing the performance of Neural Shape Compiler with CLIP-Forge (Sanghi et al., 2022) results on a simple text prompt.

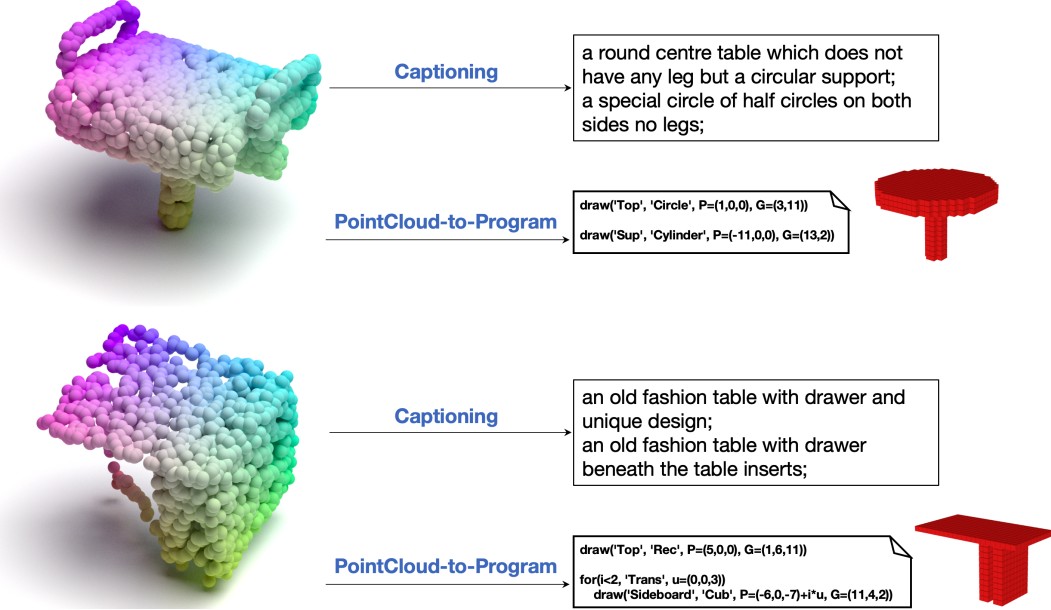

Figure 23: *Point Cloud* $\implies$ *Text* and *Point Cloud* $\implies$ *Program* results by Neural Shape Compiler given the input point clouds. One description per sentence.

