# OpenReview forum: "Neural Shape Compiler: A Unified Framework for Transforming between Text, Point Cloud, and Program"
_TMLR — Accepted by TMLR_

### Review · Reviewer_basX · 2023-01-03

**Summary Of Contributions:**

This paper presents a multimodal autoencoder for text, pointclouds, and shape-generating programs. The main idea is to encode text, pointclouds, and programs into a shared latent space, and then use domain-specific decoders to produce text, pointclouds, and programs as output. The model uses a discrete latent space like VQVAE. Like other multimodal autoencoders, it is trained with paired multimodal data, gathered from several datasets. The model benefits from the multi-modal training, and outperforms models which handle fewer modalities at once.


**Audience:**

Yes

**Claims And Evidence:**

Yes

**Requested Changes:**

It would be great to correct or disambiguate the specific sentences I pointed to, and others of the same style if possible.

**Strengths And Weaknesses:**

The main strength of this work is that it presents a straightforward and reasonable model for multi-modal inference on 3D shape data. It successfully shows the benefit of training on multi-modal data, which is not always as easy as it sounds. It also presents some interesting analysis of CLIP embeddings, and suggests plausible reasons for why CLIP-based methods fail to produce good fine-grained shape estimation (because the CLIP embeddings do not reflect fine-grained shape information).





The writing of the paper is a weak point. It sometimes uses informal or imprecise language, which often makes things ambiguous, and even confusing (at least for me). Besides typos and bad sentences, this kind of writing is problematic when it suggests something novel is happening when it's not. The first example of this is the name of the paper: "Neural Shape Compiler". There is no compiler here. The paper seems to argue that LLVM (Lattner & Adve, 2004) can be interpreted as an autoencoder, and therefore autoenders can be interpreted as compilers. This does not work for me. The LLVM paper never mentions autoencoders, and never even mentions decoders. The current work presents a fairly standard multi-modal autoencoder, where the multiple modes are sent to a shared latent space, and specialized decoders translate to target modes. The "compiled shape codes" simply refer to the output of the Transformer in the middle of the model. Much better references exist than LLVM for motivating this approach -- and several are even cited in the "Multimodal Learning" section. It feels like "compiler" is used to differentiate the approach from related work, while in fact this particular differentiation does not exist.



Here are a few more examples of ambiguous or imprecise writing.



"ShapeCode Transformer communicates with all encoders and decoders via discrete code and handles all types of transformations"



This does not seem to be true. For example it doesn't seem like any geometric transformations are handled. (How about affine? How about translating the whole object left or right?)



"In each turn T_i, we first do farthest point sampling over input point clouds"



It is unclear what is meant by a "turn" here, and it seems turns are never mentioned again.



"This paper studied how to translate between Text ⇐⇒ Point Cloud ⇐⇒ Program."

The citations preceding this quote do not do this, so I'm guessing this is supposed to refer to the current paper ("studied" should be "studies"?).



"perform compilation between the three shape abstractions with neural networks (Rumelhart et al., 1986)"



This reference seems to be mixed up. That Rumelhart paper does not involve any compilation or shape abstractions.



The paper repeatedly says it does not use positional encodings: "The decoder then combines all the 3D part codes to reconstruct whole point clouds in a permutation-equivalent manner." ... "we discard the position / coordinate information of the final points of the encoder" ... "we do not have any positional information within the source codes"

Then the paper says it uses two types of positional encodings: "Our solution is to encode our pairs of tokens with different positional embedding, i.e., we have two different positional encoding Pφtext point ([Cpoint, Ctext]) and Pφprogram point ([Cpoint, Cprogram])."

What exactly is happening here?



It seems that an important technical distinction between this work and other multi-modal autoencoders/translators is the use of a transformer at the bottleneck. The details of this module are unclear to me. What exactly are the tokens? It seems like the input and output is treated as a long sequence, and the data itself has no positional embeddings, so how does the model know when the "pointcloud" part of the sequence should end and the "text" part should begin? How many tokens are there per modality -- is it a fixed number, or does it depend on the size of the raw data?



"In training, we pad 0 at the left of data (e.g., [0, ct1 , · · · , ctNtext , · · · , cpNpoint ]) and use full attention masks"



What does a single zero contribute here? Why is this done, and why is it important?



"we turn shape programs into codes with statement types in the first place"



What does "in the first place" mean here? (Is this a specific kind of statement type?)



"Since DreamField learns a neural radiance field over a single text with tons of sampling, its training time is too long to test in the scale of our test set"



What is "tons of sampling"? What is "too long" exactly?



"Table 2" is interesting but is never referenced in the text.



"Our current framework presents some positive signals"



What exactly are positive vs negative "signals" here?



"This situation could be alleviated by the proposed of large-scale 3D dataset"



Up until this part of the paper, it did not sound like a dataset was a contribution. Maybe this sentence is from an older draft, which included a dataset contribution?



"Our current framework and data focus on correctly modeling connections between 3D geometry and text. Nevertheless, less attention has been paid to colors, textures, and materials"



I don't understand the use of "nevertheless" here. It seems like "and" would work better, which is almost opposite in meaning.

---

> ### Author Response · Authors · 2023-01-04
> **Author Response (1/4)**
>
> We greatly thank reviewer basX for the quick and thorough comments. With your help, we can contribute to a better revision.
>
>
>
> We found that there are three reasons for the above-mentioned writing problems: (1) Unskilled and improper use of English. We'd like to try our best to improve them and learn from here as well; (2) due to space limitations, some details are not included in the main text but in the appendix; (3) some of our ideas are not conveyed properly. Several passages need to be changed for better presentation.
>
>
>
> We respond to all the mentioned concerns one by one through several comments. Please let us know if some of the answers are still unclear. We've updated our manuscript with the modifications reflecting your suggestions. Thank you for your help.
>
> ---
>
>
>
> > #1: "ShapeCode Transformer communicates with all encoders and decoders via discrete code and handles all types of transformations"
> >
> > This does not seem to be true. For example it doesn't seem like any geometric transformations are handled. (How about affine? How about translating the whole object left or right?)
>
> Sorry for the confusion. In the context of this paper, "all types of transformations" refers to the transformations between different shape abstractions, i.e., $\textit{Text}$ $\Longleftrightarrow$ $\textit{Point Cloud}$ $\Longleftrightarrow$ $\textit{Program}$.
>
>
> We changed our presentation here and added an additional explanation to save the reader from thinking about other types of transformations.
>
> ---
>
>
>
> > #2: "In each turn T_i, we first do farthest point sampling over input point clouds"
> >
> > It is unclear what is meant by a "turn" here, and it seems turns are never mentioned again.
>
> Thank you for bringing this up; we've improved our presentation here in revision. Below are our responses to you on OpenReview.
>
> We originally intended to use the sentences after "In each turn $T_i$" to illustrate what is a "turn", and illustrate "turn" again in the following contents: "*we repeatedly the downsampling process (i.e., increase $T$)*."
>
> Each "turn" refers to a downsampling process happening during the encoding, including (1) farthest point sampling over input point clouds to sample a set of points noted as center points; (2) around each center point, we draw a ball with a certain radius to gather information from all points within the query ball; *[the 2nd and 4th graphs shown in the Encoder of Figure 3]* (3) we downsample the point clouds into the sampled center points with updating embeddings for each point based on the information obtained in the (2) *[the 3rd and 5th graphs shown in the Encoder of Figure 3]*. The illustrations of the encoder of Figure 3 in our paper contain 2 "turns".
>
> ---
>
>
>
> > #3: "This paper studied how to translate between Text ⇐⇒ Point Cloud ⇐⇒ Program."
> >
> > The citations preceding this quote do not do this, so I'm guessing this is supposed to refer to the current paper ("studied" should be "studies"?).
>
> Yes, you are right. It is an English error. The author of this sentence originally thought that the passive verb could represent what the paper has achieved/studied. Thanks.
>
> ---
>
> > #4: "perform compilation between the three shape abstractions with neural networks (Rumelhart et al., 1986)"
> >
> > This reference seems to be mixed up. That Rumelhart paper does not involve any compilation or shape abstractions.
>
> We included Rumelhart's paper for referring to "neural networks", not for compilation/shape abstractions. Due to the possible confusion, we've deleted the reference in the newest version.

---

> > ### Author Response · Authors · 2023-01-04
> > **Author Response (2/4)**
> >
> >
> >
> > ---
> >
> > > #5: The paper repeatedly says it does not use positional encodings: "The decoder then combines all the 3D part codes to reconstruct whole point clouds in a permutation-equivalent manner." ... "we discard the position / coordinate information of the final points of the encoder" ... "we do not have any positional information within the source codes"
> > >
> > > Then the paper says it uses two types of positional encodings: "Our solution is to encode our pairs of tokens with different positional embedding, i.e., we have two different positional encoding Pφtext point ([Cpoint, Ctext]) and Pφprogram point ([Cpoint, Cprogram])."
> > >
> > > What exactly is happening here?
> >
> > We are sorry for the confusion. Here are two types of positional information: (1) 3D coordinates for PointVQVAE encoder and decoder; (2) 1D sequence positional encoding for ShapeCode Transformer.
> >
> >
> >
> > The mentioned position/coordinate information in the first part refers to the 3D coordinates of points. For common PointCloud auto-encoders, they use 3D coordinates of points in each stage of encoding and decoding.  If you refer to Figure 3 in our paper, there are gradually down-sampled points during encoding.  Common PointCloud AEs would leverage the 3D coordinate of those points to help reconstruct. However, our framework needs to reconstruct PointCloud from Text which has no 3D coordinate information, since there is no source shape. This is one of the keys to our design and why we repeatedly say it does not use position/coordinate information in PointVQVAE. We'd like to change the term to avoid confusion with the positional encoding used in ShapeCode Transformer.
> >
> >
> >
> > The positional encoding used in ShapeCode Transformer refers to positional encoding for 1D sequence. The detailed information is included in the second paragraph of our Appendix B.2: *we have different positional embedding for each type of data pair (i.e., task), which will be optimized during the training. Since we pad 0 at the leftmost of our data, the positional embedding for the conditional code will have an extra token. For example, if we conduct ($\textit{Text}$, $\textit{Point Cloud}$) transformation, we will have $P_{text} = 256 + 1 = 257$ positional tokens for Text and $P_{point} = 128$ for Point Cloud. Therefore, we have a total of $257+128 = 385$ positional tokens for ($\textit{Text}$, $\textit{Point Cloud}$). For each positional embedding, we have the embedding dimensional 512. As a result, we have embedding size $[385, 512]$ for the task ($\textit{Text}$, $\textit{Point Cloud}$). For each task, Neural Shape Compiler has different positional embeddings but the same process described above.*
> >
> > We updated our manuscript to avoid possible confusion and misunderstanding. Thank you for bringing this up.
> >
> > ---
> >
> >
> >
> > > #6: It seems that an important technical distinction between this work and other multi-modal autoencoders/translators is the use of a transformer at the bottleneck. The details of this module are unclear to me. What exactly are the tokens? It seems like the input and output is treated as a long sequence, and the data itself has no positional embeddings, so how does the model know when the "pointcloud" part of the sequence should end and the "text" part should begin? How many tokens are there per modality -- is it a fixed number, or does it depend on the size of the raw data?
> >
> > The related information was included in Appendix B.2. We have added direct references to the appendix in the text to help illustrate.
> >
> > To answer you here in OpenReview:  Yes, you are right; the input and output are treated as a 1D-long sequence. But, we added extra positional embeddings to the sequence (as above-mention in Point #5), so the model knows when the "pointcloud" part of the sequence should end, and the "text" part should begin.
> >
> > The number of tokens is fixed. Regarding token length and vocabulary size for each shape abstraction, $\mathcal{C}^{point}$ has $N_{point} = 128$ tokens with a vocabulary size of $512$ via argmaxing from the $\textit{Point}$VQVAE encoder output; $\mathcal{C} ^{text}$ has $N_{text} = 256$ tokens of vocabulary size $49,408$; $\mathcal{C}^{program}$ has $N_{program} = 240$ tokens and a vocabulary size of $78$. Note that the number of tokens for a sentence is usually less than $N_{text} = 256$. To make all the sentences contain the same length of tokens, we pad between the last valid text token and the beginning of another type of token (e.g., program, point cloud) with values between $[49408, 49408 + 256]$.

---

> > > ### Author Response · Authors · 2023-01-04
> > > **Author Response (3/4)**
> > >
> > >
> > >
> > > ---
> > >
> > >
> > >
> > > > #7: "In training, we pad 0 at the left of data (e.g., [0, ct1 , · · · , ctNtext , · · · , cpNpoint ]) and use full attention masks"
> > > >
> > > > What does a single zero contribute here? Why is this done, and why is it important?
> > >
> > > The single zero enables our framework to perform unconditional generation. According to objective (1), we have losses starting from $k=1$. In other words, we predict tokens from the padded single zero. The padded zero can help the framework sample data unconditionally from the learned priors.
> > >
> > > We added a footnote in the method section to illustrate this design. Thanks.
> > >
> > > ---
> > >
> > >
> > >
> > > > #8: "we turn shape programs into codes with statement types in the first place"
> > > >
> > > > What does "in the first place" mean here? (Is this a specific kind of statement type?)
> > >
> > > This is probably an improper use of English. "with statement types in the first place" means we place the token that represents the statement type in the first place for the program. For example, we turn $\textit{draw(Top, Square, P=(-1,0,0), G=(2,5))}$ into $[3, -1, 0, 0, 2, 5, 0, 0]$ where $3$ represents the statement type which is the command of drawing square top, and the remains are its specific parameters. We will change this sentence in the revision.
> > >
> > >
> > >
> > > ---
> > >
> > >
> > >
> > > > #9: "Since DreamField learns a neural radiance field over a single text with tons of sampling, its training time is too long to test in the scale of our test set"
> > > >
> > > > What is "tons of sampling"? What is "too long" exactly?
> > >
> > > We included these details in Appendix B.4 point #2: we input the text prompt to DreamField by the provided command line in its GitHub for qualitative comparisons. In our environment, we need 4 GPUs of 2048 Gb memory to run a single text input with DreamField. It takes more than 8 days to finish the default 10,000 iterations for single text input. Therefore, it is impractical to test DreamField in our test set, which consists of more than 100,000 text-shape pairs.
> > >
> > >
> > >
> > > We had a reference near the main text to point to the appendix, which helped readers to check the specific time cost.
> > >
> > > ---
> > >
> > >
> > >
> > >
> > >
> > > > #10: "Table 2" is interesting but is never referenced in the text.
> > >
> > > Table 2 was referenced in the first paragraph of page 8.
> > >
> > > ---
> > >
> > >
> > >
> > > > #11: "Our current framework presents some positive signals"
> > > >
> > > > What exactly are positive vs negative "signals" here?
> > >
> > > This part is imprecise and fuzzy. We added descriptions here to make it clear in the revision and thank you for mentioning this.
> > >
> > >
> > >
> > > The positive signals in our mind are: the proposed framework has the ability to generate point clouds from input text with geometric details, generate text descriptions of the structure of input point clouds, and generate programs for composing point clouds. Compared to various baselines under multiple metrics, our jointly trained model achieved great performance in $\textit{Text}$ $\Longrightarrow$ $\textit{Point Cloud}$, $\textit{Point Cloud}$ $\Longrightarrow$ $\textit{Text}$, $\textit{Point Cloud}$ $\Longrightarrow$ $\textit{Program}$, and Point Cloud Completion tasks.
> > >
> > > ---
> > >
> > >
> > >
> > >
> > >
> > > > #12: "This situation could be alleviated by the proposed of large-scale 3D dataset"
> > > >
> > > > Up until this part of the paper, it did not sound like a dataset was a contribution. Maybe this sentence is from an older draft, which included a dataset contribution?
> > >
> > > Here are two things: (1) We have contributions for data aspects, but we do not claim them as dataset contributions. (2) The specific sentence you quote here is unrelated to our contribution but is an English error. We extend the two points below.
> > >
> > >
> > >
> > > (1) We have made non-trivial efforts to adjust and annotate data from Text2Shape, ShapeGlot, and ABO datasets, resulting in 107,371 ($\textit{Point Cloud}$, $\textit{Structure-Related Text}$) pairs. The related details are included in Appendix A.2. To better answer here in Openreview, we include a brief description of what we have done: (1) removing descriptions around artificial colors, textures, and materials in Text2Shape dataset; (2) turning triple relationships into binary text-shape pairs for ShapeGlot dataset; (3) removing geometry-non-related words (e.g., brand names) and annotate text for ABO dataset.
> > >
> > >
> > >
> > > (2) for the specific sentence, we find it is better to re-statement as "This situation could be alleviated by the development of large-scale 3D datasets in the future". It is an inappropriate use of English. We did not intend to claim we have a dataset contribution that can alleviate this situation.
> > >
> > > ---
> > >
> > >
> > >
> > > > #13: "Our current framework and data focus on correctly modeling connections between 3D geometry and text. Nevertheless, less attention has been paid to colors, textures, and materials"
> > > >
> > > > I don't understand the use of "nevertheless" here. It seems like "and" would work better, which is almost opposite in meaning.
> > >
> > > This is an English error. We fixed it. Thanks.

---

> ### Author Response · Authors · 2023-01-04
> **Author Response (4/4)**
>
> ---
>
> > #0: The first example of this is the name of the paper: "Neural Shape Compiler". There is no compiler here. The paper seems to argue that LLVM (Lattner & Adve, 2004) can be interpreted as an autoencoder, and therefore autoenders can be interpreted as compilers. This does not work for me. The LLVM paper never mentions autoencoders, and never even mentions decoders. The current work presents a fairly standard multi-modal autoencoder, where the multiple modes are sent to a shared latent space, and specialized decoders translate to target modes. The "compiled shape codes" simply refer to the output of the Transformer in the middle of the model. Much better references exist than LLVM for motivating this approach -- and several are even cited in the "Multimodal Learning" section. It feels like "compiler" is used to differentiate the approach from related work, while in fact this particular differentiation does not exist.
>
>
>
> We believe this is perhaps one of the most important concerns, and share our thoughts with you via this comment seriously. We'd like to best present our paper in a precise and not overly exaggerated manner, and thank you very much for your help on this point.
>
>
>
> We are indeed motivated by the concept of modern compilers: this paper's idea came up when we coincidentally took a glimpse at this article https://cacm.acm.org/magazines/2022/2/258231-abstractions-their-algorithms-and-their-compilers/fulltext. We are motivated by the high-level structure of compilers: they turn source codes into intermediate representations (IRs), transform IRs, and decode them into other types of high-level programming languages, as shown in the first Figure of https://blog.gopheracademy.com/advent-2018/llvm-ir-and-go/ and Figure 3.4 in https://livebook.manning.com/book/webassembly-in-action/chapter-3/53.
>
>
>
> We understand there are fundamental differences between the program compiler and our framework. Therefore, we emphasized the difference in Section 2: (1) *compilers translate high-level program language (e.g., Pascal, C) into executable machine code with the help of assemblers and linkers*; (2) *A major difference between our framework and modern program compilers is that our IR transformation process is probabilistic, whereas the process in a program compiler is a deterministic process with potential performance optimizations.*
>
>
>
> Also, we set a further goal for our framework in Section 5: *Neural Shape Compiler currently compiles different shape abstractions. Compared to this, a further direction is to compile operations on one shape abstraction into practical steps in other abstractions and change it accordingly. For example, we add "$\textit{armless}$" into a text description of a chair with two arms. This change can be effectively compiled into operations "\textit{remove two arm parts and their symmetry relationship}" in the shape hierarchy and "$\textit{delete all points belonging to the two arms}$" in point clouds. We leave the shape editing direction to future study.*
>
>
>
> To directly answer the specific questions: we are not trying to (1) argue LLVM can be interpreted as an autoencoder, and therefore autoencoders can be interpreted as compilers; (2) use "compiler" to differentiate the approach from related work. At the beginning of the paper, we directly point out why the program compiler motivates this work without claiming our framework serves similar functions as program compilers do. We also point out the difference to people who may not be that familiar with LLVM and Emscripten.
>
> As you mentioned, we included the related multimodal learning papers in the related work section, but, they are not directly motivating our idea. Here's the situation: we developed approach ***A***, which is motivated by the works of direction ***B***, but some papers ***C*** also look similar to ***A***. The solution we currently chose is to mention the direct motivation source ***A*** in the introduction but also make the difference clear, and mention ***C*** in related works. We welcome any suggestions here.

---

> > ### Comment · Reviewer_basX · 2023-01-31
> > **Thank you**
> >
> > The revisions listed here make the paper much better in my view. I might come back and pick on a few points, but overall I am happy to see so much progress.
> >
> > I am still not convinced about calling the method a "compiler". It makes sense that the referenced compiler papers provided some inspiration, and this is interesting and good to write about, but this does not make "compiler" a valid descriptor of the method. Part of what makes this problematic (or at least confusing) is that the technique presented here already has (I think) a standard name, which is "multimodal autoencoder". A good reference on this might be Ngiam et al., "Multimodal Deep Learning", ICML 2011. Figure 3-b in that work shows the same key idea as Figure 2-left in the current draft.

---

> > > ### Author Response · Authors · 2023-02-01
> > > **Author Response**
> > >
> > > Thank you for reviewing and acknowledging our revision efforts. We welcome any further feedback.
> > >
> > > We are willing to seriously discuss and communicate with you on the "compiler" aspects until we finally reach an agreement. Below are our further thoughts about "multimodal autoencoder" and "compiler":
> > >
> > > (1) Figure 2-left in our paper is indeed similar to Figure 3-b in [Ngiam et al., ICML11]. However, the key difference is our ShapeCodes aren't shared with each other. The ShapeCodes are different across different shape representations. Specifically, $\mathcal{C}^{point}$ has $N_{point} = 128$ tokens with a vocabulary size of $512$; $\mathcal{C} ^{text}$ has $N_{text} = 256$ tokens of vocabulary size $49,408$; $\mathcal{C}^{program}$ has $N_{program} = 240$ tokens and a vocabulary size of $78$.
> > >
> > > Our ShapeCodes are different from each other, but all in a unified discrete space, making the ShapeCode Transformer works along with different representations. Therefore, directly using "multimodal autoencoder" may confuse people about whether our framework adopted "shared representations".
> > >
> > > (2) According to [Wikipedia](https://en.wikipedia.org/wiki/Compiler) and its references [Compilers: principles, techniques, & tools, Aho et al., 2007], which is also cited in our paper, "a compiler is a computer program that translates computer code written in one programming language (the source language) into another language (the target language)."
> > >
> > > As for our framework, we translate shape representation written in one ShapeCode (corresponding to the source representation) into another type of ShapeCode (corresponding to the target representation) with neural networks. This is analogical to general ideas of "compiler". The name "Neural Shape Compiler" indicates (1) performing transformations between different shape representations; and (2) using neural networks: they are consistent with what actually happens.
> > >
> > > Also, as stated in [Author Response (4/4)](https://openreview.net/forum?id=gR9UVgH8PZ&noteId=e1CWeNpSEE), we have emphasized the difference between our framework with program compilers in several places of our paper. To further clarify this point and prevent any misunderstanding, ***we put a footnote on the first page to emphasize our framework has fundamental differences from program compilers.***
> > >
> > > We welcome any further thoughts about this point.

---

### Review · Reviewer_eY29 · 2023-01-07

**Summary Of Contributions:**

This paper builds a unified framework, called Neural Shape Compiler, to transform between shape abstractions: Text, Point Cloud, and Programs.
This framework includes a PointVQVAE (as the encoder and the decoder), and also a ShapeCode Transformer.
On diverse benchmarks, including Text2Shape, ShapeGlot, ABO, Genre, and Program Synthetic datasets, the proposed Neural Shape Compiler shows better performances.

**Audience:**

Yes

**Broader Impact Concerns:**

To better show the broader impact of this work, the authors could consider explaining more on: 1) why it is necessary to build a unified framework to translate between pairs of shape abstractions; 2) why previous work failed in this unified framework.

**Claims And Evidence:**

Yes

**Requested Changes:**

Could the authors more clearly elaborate on:
1. The motivations and the unique contributions of your model design (PointVQVAE and ShapeCode Transformer)? Especially, compare with previous works: VQVAE, PointNet++, and Ramesh et al., 2021.
2. Why do the authors follow the use of Chamfer distance (Fan et al., 2017) and Earth Mover’s distance (Rubner et al., 2000)?

I believe Section 3 could be better motivated and organized, so that readers could follow the motivation why the authors choose these options.

**Strengths And Weaknesses:**

I have to clarify that this paper is completely out of my expertise. I never published any paper in the 3D vision domain.

## Strengths:
The work proposed a versatile framework to transform between point cloud, text, and programs. The authors conducted comprehensive experiments with clearly elaborated settings and comparisons.

## Weaknesses
1. Any quantitive ablation study on the number of downsampling processes and choices of $R_{T_i}$ in section 3.1?
2. Are all results in experiments compared with fair model sizes and computation costs (FLOPs)?

---

> ### Author Response · Authors · 2023-01-08
> **Author Response (1/2)**
>
> The authors are grateful to reviewer eY29 for the valuable feedback. Also, we appreciate the clarification of expertise, which will assist the action editor in making the final decision. Your comments are invaluable as we revise the draft to suit general readers better.
>
>
>
> In the following, we address your concerns on OpenReview. We have integrated some changes into our revision.
>
>
>
> ---
>
> > #1: To better show the broader impact of this work, the authors could consider explaining more on: why it is necessary to build a unified framework to translate between pairs of shape abstractions?
>
> Thanks for helping us present better. We improved our presentation to motivate our framework better in the revision.
>
> To answer you in OpenReview:
>
>
> There have been too many different representations for 3D shapes, such as PointCloud, Mesh, Voxel, Implicit Functions, Multi-View Images, Program, CSG Tree, Hierarchy, Graph, Text, etc. Researchers chose different representations and designed specialized models for different tasks. We believe this is not an optimal way to solve 3D tasks.
>
>
>
> One lesson we can learn from LLM is that it is more effective to focus on a single, flexible architecture that can adapt to various tasks rather than using specialized architectures for each task. The only difference between tasks should be the data, not the model itself. This paper proposed a framework to take a step further in this direction. The performance difference between *Shape Compiler* and *Shape Compiler Limited* in Tables 1, 3, 4, and 5 demonstrates combining the learning of different transformation branches within a unified framework leads to stronger representation learning and improved performance on each individual task.
>
>
>
> Besides, we believe it is important to form a [Bridge](https://drive.google.com/file/d/1oVeZkc5TPuPSzcOTduGAd_RmAAewlLQn/view?usp=sharing) across different shape representations and provide complete information on 3D shapes for general purposes, including geometrical and compositional information, regularities, and semantic meanings ([Figure](https://drive.google.com/file/d/1vZkogwGtG4i0wfsK14FUmewllEhloNHq/view)).
>
> ---
>
>
>
> > #2: The motivations and the unique contributions of your model design (PointVQVAE and ShapeCode Transformer)? Especially, compare with previous works: VQVAE, PointNet++, and Ramesh et al., 2021.
> >
> > why previous work failed in this unified framework?
>
> The key in our unified framework is to adopt the unified discrete Shape Code as the intermediate representation for different shape representations. Then, ShapeCode Transformer can learn and optimize for all different tasks in a unified space with a single objective (Equation 1). The motivation and contribution of PointVQVAE and ShapeCode Transformer all come from the final unified framework.
>
>
>
> In other words, PointVQVAE turns PointClouds into discrete codes. Then, the proposed ShapeCode Translator can act in the unified space, and the joint training over different shape representations comes true. They all contribute to the improvements/achievements in the final tasks: $\textit{Text}$ $\Longrightarrow$ $\textit{Point Cloud}$, $\textit{Point Cloud}$ $\Longrightarrow$ $\textit{Text}$, $\textit{Point Cloud}$ $\Longrightarrow$ $\textit{Program}$, and Point Cloud Completion.
>
>
>
> Regarding specific previous papers: (1) VQVAE are designed to reconstruct images from code, which directly motivates our PointVQVAE and cannot process Point Cloud without our designs; (2) PointNet++ can process Point Clouds with continuous representation, which cannot be fitted into our unified framework; (3) our framework extends Ramesh et al., 2021 to process multiple pairs of transformation. Note that, for the sake of generality, both our PointVQVAE and ShapeCode transformer is implemented as the most standard architecture, rather than other variants.
>
>
>
> We integrated the above discussion in our revision to show the difference with previous works better. Thank you for bringing this up.
>
> ---
> > #3: Section 3 could be better motivated and organized, so that readers could follow the motivation why the authors choose these options.
> >
> Thank you for your suggestion. We re-wrote many parts of Section 3 and made things more organized and well-motivated.

---

> > ### Author Response · Authors · 2023-01-08
> > **Author Response (2/2)**
> >
> >
> >
> > ---
> >
> >
> >
> > > #4: Why do the authors follow the use of Chamfer distance (Fan et al., 2017) and Earth Mover’s distance (Rubner et al., 2000)?
> >
> > Chamfer distance and Earth Mover’s distance are used to supervise the reconstruction training of PointVQVAE. During training, both the input and output of PointVQVAE are Point Clouds. It is common to introduce Chamfer distance and Earth Mover’s distance to compute the gradient effectively between two Point Clouds, i.e., the original Point Clouds and the reconstructed Point Clouds from ShapeCode in our case. Some further details are included in Appendix B.1.
> >
> > ---
> >
> >
> >
> > > #5: Any quantitive ablation study on the number of downsampling processes and choices of $R_{T_i}$ in section 3.1?
> >
> > For PointVQVAE, one of the most important ablation studies is setting $R_{T_i}$ to a very large value that enables the final output of the encoder to have the global receptive field. The experimental result ("PointVQVAE globalpool") in Table 6 shows setting a very large $R$ lead to drastic performance drops. Another important factor is how many points we sampled in the last layer of the encoder. We tested five settings (16, 32, 64, 128, 256) and found more points are consistently better. Finally, we chose 64 as a balanced value (not the best performance one, 256) in our framework.
> >
> > For the sake of generality and computational cost, we did not exhaustively search the hyperparameters. We've tested several ablations about $R_{T_i}$ as shown below, including both two and three times of downsampling processes. In our experiments, we chose [$R_{T_0} = 0.1$, $R_{T_1}=0.4$] due to its good performance and light weight.
> >
> >
> > | Configurations                                   | Reconstruction Performance on the Validation set (mean CD $\downarrow$) |
> > | ------------------------------------------------ | ------------------------------------------------------------ |
> > | [$R_{T_1} = 0.1$, $R_{T_2}=0.2$]                 | 0.013356                                                     |
> > | [$R_{T_1} = 0.1$, $R_{T_2}=0.4$]                 | 0.006877                                                     |
> > | [$R_{T_1} = 0.1$, $R_{T_2}=0.6$]                 | 0.008283                                                     |
> > | [$R_{T_1} = 0.05$, $R_{T_2}=0.1$, $R_{T_3}=0.6$] | 0.006513                                                     |
> > | [$R_{T_1} = 0.05$, $R_{T_2}=0.2$, $R_{T_3}=0.6$] | 0.007808                                                     |
> >
> >
> >
> >
> > ----
> >
> >
> >
> > > #6: Are all results in experiments compared with fair model sizes and computation costs (FLOPs)?
> >
> > Most of our baselines consist of two stages, the first stage encodes shapes and texts into embeddings, and the second stage learns the target embeddings by executing the learned generative model. Therefore, we count the model parameters for the encoder and generative model separately. The only exception is Dream Fields which mainly consists of a generative model consisting of a small NeRF and a CLIP model (we used ViT-B/16). Since all methods except Dream Fields do not use the gradient of the pretrained text encoder, we do not consider the text encoder parameters for all methods.
> >
> > Some statistics are shown below. Since we compare methods from 2018 to nowadays, it is expected we will have very different model sizes across different methods.
> >
> > | Methods        | Encoder Parameters | Generative Model Parameters |
> > | -------------- | ------------------ | --------------------------- |
> > | Dream Fields   | -                  | 149,691,777                 |
> > | CWGAN          | 4,183,618          | 44,101,313                  |
> > | Shape IMLE     | 11,149,824         | 117,534,920                 |
> > | CLIP-Forge     | 5,409,665          | 18,373,120                  |
> > | Shape Compiler | 26,795,008         | 36,741,120                  |

---

### Review · Reviewer_qL43 · 2023-01-09

**Summary Of Contributions:**

The paper proposes a multi-modal auto-encoder model for 3D shapes.

Specifically, the authors propose a set of VQ-encoders to encode point clouds, programs, and text descriptions into three different kinds of VQ shape codes.

The authors then propose a ShapeTransformer to transform one kind of latent code into another kind of latent code.

Three different decoders then decode the domain-specific latent codes into the appropriate representation.

The model is trained end-to-end, and the authors demonstrate not only that training on multi-modal data improves performance over training a single-domain auto-encoder, but also demonstrate significant qualitative and quantitative gains over a variety of baselines.

**Audience:**

Yes

**Broader Impact Concerns:**

No concerns on ethical implications.

**Claims And Evidence:**

Yes

**Requested Changes:**

## "Unified" shape codes
On the one hand side, the paper again and again highlights "unified shape codes", suggesting that one shape code can be decoded into all three representations. It turns out that this is *not* what unified means - in fact, I'm very confused about what "unified" is supposed to mean here. Instead, the paper talks about the "ShapeCode Transformer" as the model that *converts* one shape code into another. This is already confusing in the abstract, and then, Fig. 2 is confusing yet again: The different front-ends are all encoded into a "ShapeCode", but then, the "ShapeCode Transformer" outputs yet another "ShapeCode". Both the first and the second ShapeCode in Fig. 2 are suggested to be domain-agnostic (i.e., the same ShapeCode is used for all of Text, Point Cloud, and Program), but the text suggests "ShapeCode Transformer ... transforms shape codes of one type into another". Later, in 3.2, it's clarified that the ShapeCodes *are* different across different representations.

1. The use of "unified" should be removed.
2. Fig. 2 should visualize the three different kinds of shape codes, and the transformer should translate between them.

## Bold numbers in tables
In table 1, 3, the best-performing method should be bolded.

## Misc
Page 2: permutation-equivalent --> permutation equivariant?
4.1. "Tons of sampling" --> "A lot of sampling".

**Strengths And Weaknesses:**

## Strengths
+ The paper clearly argues a core hypothesis, namely that training a joint encoder-decoder architecture across different modalities improves on shape reconstruction & generation tasks.
+ The model is well-motivated and well-crafted; The use of a vector-quantized latent space makes sense, the architecture of the probabilistic ShapeTransformer makes sense.
+ Results are convincing, with cool generations and a clear improvement over baselines.
+ I find the investigation into the properties of CLIP embeddings in Table 2 well-motivated and insightful.
+ Great related work section.
+ I find the discussion of limitations great. It is absolutely expected that the proposed method cannot appropriately describe the hilbert cube, but good to see and to introduce as a reasonable task. The piontcloud-2-program result is still convincing.
+ Very thorough evaluation.

## Weaknesses
- I find that the grammar is a bit lacking from time to time - not a huge deal, but readibility could be improved.
- I find that the notion of "unified latent space" is extremely confusing, please see "Requested Changes".
- I find the structure of section 3.1 a bit lacking. I think it would be clearer to move some of the details on the point cloud decoder into this section, and then to have one paragraph per encoder-decoder type.

All in all, I believe that this paper is a valuable contribution to the field and a valuable reference.

---

> ### Author Response · Authors · 2023-01-12
> **Author Response**
>
> We greatly appreciate Reviewer qL43 for the comprehensive reviews. It is gratified for us that receive your recognition for the many efforts we have put into our manuscript. We'll contribute to a better revision with your help here.
>
> We respond to your concerns below on OpenReview. Some of the changes are integrated into our revised manuscript.
>
> ---
> > I find that the grammar is a bit lacking from time to time - not a huge deal, but readibility could be improved.
>
> This problem was mainly caused by our unskilled use of English. We'd try our best to improve our use of English, which currently lacks proficiency or accuracy, and to learn from this experience.
>
>
> ---
> > I find that the notion of "unified latent space" is extremely confusing.
> >
> > On the one hand side, the paper again and again highlights "unified shape codes", suggesting that one shape code can be decoded into all three representations. It turns out that this is *not* what unified means - in fact, I'm very confused about what "unified" is supposed to mean here. Instead, the paper talks about the "ShapeCode Transformer" as the model that *converts* one shape code into another. This is already confusing in the abstract, and then, Fig. 2 is confusing yet again: The different front-ends are all encoded into a "ShapeCode", but then, the "ShapeCode Transformer" outputs yet another "ShapeCode". Both the first and the second ShapeCode in Fig. 2 are suggested to be domain-agnostic (i.e., the same ShapeCode is used for all of Text, Point Cloud, and Program), but the text suggests "ShapeCode Transformer ... transforms shape codes of one type into another". Later, in 3.2, it's clarified that the ShapeCodes *are* different across different representations.
>
> Sorry for the confusion, and thank you for bringing this up. Based on your feedback, we are aware that the term "unified shape codes" can be misleading to think that our shape codes are domain-agnostic for different representations. As a fix, we deleted "unified" before every shape code and changed the terminology to unified discrete space.
>
> Your understanding is correct; the ShapeCodes are different across different representations. Specifically, $\mathcal{C}^{point}$ has $N_{point} = 128$ tokens with a vocabulary size of $512$; $\mathcal{C} ^{text}$ has $N_{text} = 256$ tokens of vocabulary size $49,408$; $\mathcal{C}^{program}$ has $N_{program} = 240$ tokens and a vocabulary size of $78$.
>
> The reason why we use unified shape codes in our current manuscript is all types of ShapeCodes are discrete codes. In other words, Text, point clouds, and programs have very different representations from each other. But, we turn the three different representations into the unified discrete representation, and then, our ShapeCode Transformer can jointly learn over all heterogeneous tasks. Therefore, in our current version, we highlighted this design at the beginning of the paper, and then illustrated specific differences in the methods and implementation sections. In this way, we hope readers can appreciate our design while knowing the detailed implementation as well.
>
>
>
>
> ---
> > I find the structure of section 3.1 a bit lacking. I think it would be clearer to move some of the details on the point cloud decoder into this section, and then to have one paragraph per encoder-decoder type.
>
> Thank you for your advice. We took your suggestion seriously and changed our presentation in Section 3.1 to make things more structured.
>
> ---
> > In table 1, 3, the best-performing method should be bolded.
> >
> > Page 2: permutation-equivalent --> permutation equivariant? 4.1. "Tons of sampling" --> "A lot of sampling".
>
> Our updated revision has corrected the above-mentioned issues. Thank you!

---

### Decision · Action_Editors · 2023-03-02

**Recommendation:** Accept as is

**Comment:**

The reviewers found the paper's claims to be clearly made and well-supported by evidence. The proposed architecture is not especially surprising or novel. However, the evaluation is thorough, and the results should be of interest to a significant segment of the ML community. The authors also improved the paper significantly during the revision process, and the reviewers appreciated this.

Given all this, I am recommending that the paper be accepted in its current form.

**Audience:**

The paper's results should be of interest to researchers working on 3D modeling, as well as the increasingly important problem of multimodal inference.

**Claims And Evidence:**

The paper gives a framework for multimodal inference on 3D shape data. The process has three parts. First, a set of distinct encoders are used to encode the different kinds of input data (in this case, point clouds, programs, and textual descriptions) into different types of latent codes. Second, a ShapeTransformer is used to go back and forth between the different latent representations. Third, three distinct decoders are used to map the latent representations back to data of the appropriate type. The empirical evaluation is thorough. It shows that the framework can solve complex inference tasks, demonstrates the benefit of joint training on multi-modal data, and shows some interesting analysis of CLIP embeddings.